# GenoArmory: A Unified Evaluation Framework for Adversarial Attacks on Genomic Foundation Models

## Abstract

We propose the **first** unified adversarial attack benchmark for Genomic Foundation Models (GFMs), named **GenoArmory**. Unlike existing GFM benchmarks, GenoArmory offers the first comprehensive evaluation framework to systematically assess the vulnerability of GFMs to adversarial attacks. Methodologically, we evaluate the adversarial robustness of five state-of-the-art GFMs using four widely adopted attack algorithms and three defense strategies. Importantly, our benchmark provides an accessible and comprehensive framework to analyze GFM vulnerabilities with respect to model architecture, quantization schemes, and training datasets. Additionally, we introduce **GenoAdv**, a new adversarial sample dataset designed to improve GFM safety. Empirically, classification models exhibit greater robustness to adversarial perturbations compared to generative models, highlighting the impact of task type on model vulnerability. Moreover, adversarial attacks frequently target biologically significant genomic regions, suggesting that these models effectively capture meaningful sequence features.

## 1 Introduction

The advent of Genomic Foundation Models (GFMs) has revolutionized the analysis and generation of DNA and RNA sequences (Zhou et al., 2025b;a; 2024; Ye et al., 2024; Nguyen et al., 2024a; Dalla-Torre et al., 2024; Nguyen et al., 2024b; Ji et al., 2021). These models, pre-trained on extensive genomic datasets, have demonstrated exceptional performance across a variety of genomics tasks, leading to widespread adoption in both research and industry. For instance, GFMs have shown proficiency in generating high-quality DNA and RNA sequences (Zhou et al., 2025b; Nguyen et al., 2024a) and in species classification tasks (Zhou et al., 2024; Dalla-Torre et al., 2024; Ji et al., 2021). In the realm of medical diagnostics, GFMs contribute significantly by predicting gene pathogenicity (Sayeed et al., 2024) and assessing genome-wide variant effects (Benegas et al., 2023). Their capabilities extend to functional genomics, aiding in promoter detection (Fishman et al., 2025) and transcription factor prediction (Fu et al., 2025; Kabir et al., 2024), which are crucial for understanding gene regulation mechanisms. GFMs also are instrumental in RNA secondary structure prediction (Yang & Li, 2024), a critical aspect of understanding RNA function and interactions.

Despite the remarkable advancements, GFMs face significant challenges, particularly concerning their robustness and security. GFMs, which process structured, high-dimensional, and low-redundancy inputs like DNA sequences, are especially susceptible to adversarial attacks—even minor perturbations, such as single-nucleotide variations, can lead to substantial biological consequences. For instance, recent studies (Montserrat & Ioannidis, 2023) have demonstrated that DNA language models, including DNABERT-2 and the Nucleotide Transformer, are vulnerable to various adversarial strategies including nucleotide-level substitutions, codon-level modifications, and backtranslation-based transformations. Such attacks can significantly degrade model performance in tasks like antimicrobial resistance gene classification and promoter detection. Moreover, the generative capabilities of GFMs can be exploited by the attacker—it could manipulate models like GenomeOcean (Zhou et al., 2025b) to produce biologically nonsensical sequences, potentially leading to harmful application, even including the design of bioweapons (Peppin et al., 2024).

Figure 1: **An overview of benchmarking adversarial attacks on GFMs**

Given the significant safety concerns surrounding GFMs, there is a pressing need for robust defense mechanisms to ensure their reliability and security. However, the absence of benchmarks specifically designed to evaluate GFM safety has hindered the development of effective defense methods. Existing efforts (Zhou et al., 2024; Liu et al., 2025) primarily assess performance, without addressing safety aspects. This highlights the urgency of developing a new benchmark specifically designed to evaluate the safety of GFMs. To address this need, we introduce the GenoArmory benchmark, as shown in Figure 1, designed to standardize best practices in the emerging field of adversarial attack and defense for DNA-based GFMs. GenoArmory is guided by core principles of transparency, reproducibility, and fairness in evaluating GFM robustness under both attack and defense scenarios. In this paper, we detail these guiding principles, describe the benchmark's components, report results across multiple attack and defense strategies on various GFMs, and share insights to inform robustness improvements.

**Contributions:** We propose the GenoArmory framework (Figure 2) to a comprehensively assess the robustness of GFMs against adversarial attacks. Our contributions include:

- **Pipeline for red-teaming GFMs.** We present a comprehensive evaluation pipeline to assess the robustness of DNA-based GFMs against adversarial attacks. Specifically, our pipeline implements both gradient-based and gradient-free attack strategies across five different GFMs with standardized evaluation metrics.

- **Pipeline for testing and adding new defenses.** We implement three defense mechanisms and evaluate their effectiveness against adversarial attacks. Additionally, we provide plug-and-play code to enable standardized evaluation of newly developed defense methods.

- **Repository of GFM adversarial attack artifacts.** We provide a repository of adversarial attack artifacts on GFMs, including adversarial examples and attack code, to facilitate reproducibility and further research in this area.

- **New adversarial sample dataset for GFMs.** We introduce a new dataset **GenoAdv**, composed of adversarial examples specifically generated to improve the robustness of GFMs. When used in training, GenoAdv yield a **34.71%** Defense Success Rate, compared to training using only TextFooler samples.

- **Meaningful insights.** We provide a comprehensive analysis of GFM robustness under adversarial attacks, revealing the strengths and limitations of various models and defense strategies. Additionally, we offer an in-depth discussion on how training methods and quantization settings impact the robustness of GFMs.

## 2 BACKGROUND

**Definition.** Given a genomic sequence $X = [x_1, x_2, \ldots, x_n]$, where each nucleotide $x_i \in \{A, T, C, G\}$, a DNA model $f(\cdot)$, and a corresponding label $y$, our goal is to find an adversarial sequence $X'$ that satisfies:

$$f(X') \neq y \quad \text{subject to} \quad d(X, X') \leq \epsilon,$$

where $d(\cdot, \cdot)$ is a distance metric measuring the perturbation between the original and adversarial sequences, and $\epsilon$ controls the perturbation budget.

**Genomic Foundation Models.** Recent advances in genomic foundation models (GFMs) (Liu et al., 2025) establish two principal methodological paradigms: classification models and generative models. Within the classification paradigm, transformer-based approaches exhibit progressive technical

refinements. Initial models, including DNABERT (Ji et al., 2021) and Nucleotide Transformer (Dalla-Torre et al., 2024), establish baseline performance through fixed k-mer tokenization strategies. DNABERT-2 (Zhou et al., 2024) addresses these constraints by integrating byte-pair encoding (BPE) for tokenization and Attention with Linear Biases (ALiBi) for modeling longer sequences, which significantly enhances motif discovery capabilities. Building on this, DNABERT-S (Zhou et al., 2025a) focuses on species differences in the embedding space. GERM (Luo et al., 2025) emerges as the first GFM specifically optimized for resource-constrained environments. By integrating an outlier-free architecture, GERM achieves both reliable quantization and fast adaptation. For long-range genomic dependency modeling, HyenaDNA (Nguyen et al., 2024b) replaces conventional attention mechanisms with Hyena operators, enabling efficient processing of ultra-long genomic sequences. Among generative models, GenomeOcean (Zhou et al., 2025b) represents a pioneer, trains on 220TB of genomic data, and demonstrates strong DNA sequence generation capabilities across diverse species domains. Meanwhile, Evo (Nguyen et al., 2024a) introduces a hybrid architecture that combines Hyena operators with sparse attention mechanisms capable of performing whole-genome modeling at single nucleotide resolution.

**Attack Methods.** As shown in Figure 5, adversarial attacks are broadly categorized into untargeted, targeted, and universal variants. Untargeted attacks (Liu et al., 2019b; Madry et al., 2018a) aim to maximize model loss by perturbing inputs toward the gradient, while targeted attacks (Carlini & Wagner, 2017; Zhang et al., 2024) steer predictions toward specific classes by gradient. Universal attacks (Moosavi-Dezfooli et al., 2017) generate input-agnostic perturbations that mislead models across entire data distributions. Numerous adversarial attack methods have been proposed in both NLP and CV, demonstrating their effectiveness in impacting model performance. Only one work, FIMBA (Skovorodnikov & Alkhzaimi, 2024), propose adversarial attacks in the genomic domain. FIMBA introduces a black-box, model-agnostic framework that perturbs key features identified via SHAP values to disrupt genomic models.

**Defense Methods.** As shown in Figure 5, defense strategies are broadly categorized into adversarial training, defensive distillation, adversarial sample detection, and regularization with certified robustness. Adversarial training (Zhu et al., 2020; Madry et al., 2018a) enhances model robustness by iteratively injecting adversarial examples during training, Another approach defensive distillation (Papernot et al., 2016) trains student models on softened probability distributions from teacher models to smooth decision boundaries. In contrast, adversarial sample (Jin et al., 2024; Zheng et al., 2023b; Qi et al., 2021) detection identifies malicious inputs at inference time. Regularization with certified robustness (Li et al., 2023; Liu et al., 2022; Ye et al., 2020; Jia et al., 2019) reduces vulnerability through loss shaping.

## 3 MAIN FEATURES FOR GENOARMORY

Given the current landscape of GFMs, there exists no benchmark dedicated to evaluating their reliability. Considering the significant safety concerns, we propose the **first** benchmark, **GenoArmory**, targeting adversarial attacks—one of the most critical threats to GFM security. GenoArmory supports state-of-the-art attacks and defenses on GFMs, as well as providing direct access to the corresponding adversarial attack artifacts. In particular, we prioritize the following aspects in our benchmark: It will continuously update to incorporate emerging attacks and defenses from the literature. Additionally, we aim to evolve the benchmark alongside the community to support newly developed methods.

### 3.1 GENOADV: A DATASET OF ADVERSARIAL EXAMPLES ON GFMS

An important contribution of this work is the creation of an adversarial example dataset for GFMs, named **GenoAdv**. This dataset comprises adversarial examples generated using multiple attack methods—BertAttack (Li et al., 2020a), TextFooler (Jin et al., 2020), and FIMBA (Skovorodnikov & Alkhzaimi, 2024)—on various GFMs. While prior studies (Li et al., 2020c; Zheng et al., 2020; Liu et al., 2019a) leverage transferable adversarial examples for training, the effectiveness of such transferability remains questionable. To address this, we generate adversarial examples using diverse techniques to better capture model-specific vulnerabilities. The GenoAdv dataset offers a comprehensive and diverse set of adversarial examples across different tasks and methods, providing users with a practical resource for rapid adversarial training to enhance model robustness.

Figure 2: **GenoArmory Framework.** Our GenoArmory framework incorporates diverse adversarial attack and defense methods on GFMs. It also offers visualization tools to highlight important regions influencing model predictions and introduces a new adversarial dataset, **GenoAdv**.

## 3.2 A REPOSITORY OF ADVERSARIAL ATTACKS ARTIFACTS

A central component of the GenoArmory benchmark is our accessible repository of adversarial attack artifacts. Given the limited availability of GFM-specific adversarial attack method—FIMBA (Skovorodnikov & Alkhzaimi, 2024) being the only one to date—we adapt existing attack techniques from language and computer vision domains to GFMs. As a result, the GenoArmory artifact repository includes adversarial examples generated by BertAttack (Li et al., 2020a), TextFooler (Jin et al., 2020), PGD (Madry et al., 2018b), and FIMBA (Skovorodnikov & Alkhzaimi, 2024).

```
from GenoArmory import GenoArmory
gen = GenoArmory(model="DNABERT-2-finetuned-H3", tokenizer="DNABERT-2-finetuned-H3")
gen.get_attack_metadata(method=TextFooler,model_name=dnabert)
```

## 3.3 A PIPELINE FOR RED-TEAMING GFMS

Adversarial attacks on GFMs are challenging due to variations in tokenization, architecture, and datasets, leading to inconsistent results. To address this, we propose a standardized red-teaming pipeline that includes pre-trained GFMs, datasets, hyperparameters, and adversarial examples. The pipeline integrates five state-of-the-art models—DNABERT-2 (Zhou et al., 2024), Nucleotide Transformer (NT, NT2) (Dalla-Torre et al., 2024), GenomeOcean (Zhou et al., 2025b), and HyenaDNA (Nguyen et al., 2024b)—along with 26 DNA-based classification datasets. It provides direct access to attack artifacts section 3.2 for standardized evaluation of adversarial robustness and supports user-defined attack methods, offering a flexible and extensible framework for evaluating model robustness.

```
import json
with open(params_file, "r") as f:
    kwargs = json.load(f)
gen.attack(attack_method='pgd', **kwargs)
```

## 3.4 A PIPELINE FOR EVALUATING DEFENSES AGAINST ADVERSARIAL ATTACKS

In addition to efforts in developing new attack methods, researchers propose various defense strategies to counter adversarial threats. Our benchmark provides a standardized pipeline for evaluating the effectiveness of these defenses against adversarial attacks. Since no defense methods have been specifically designed for GFMs, we adapt existing state-of-the-arts from natural language and computer vision domains, i.e., adversarial training (Zheng et al., 2020), ADFAR (Bao et al., 2021), and FreeLB (Zhu et al., 2020), as defense baselines for GFMs. In our evaluation, we adopt existing attack methods as the base and assess the robustness of the defenses against adversarial examples generated by these attacks.

```
gen.defense(defense_method='freelb', **kwargs)
```

## 3.5 REPRODUCIBLE EVALUATION FRAMEWORK

In addition to providing access to the attack artifacts and defense strategies, we present a standardized evaluation framework, enabling users to benchmark robustness methods. The framework includes all essential components—data loading, model training and evaluation, and accuracy-based metrics. A detailed discussion on reproducibility is provided in appendix F.

## 3.6 A LIGHTWEIGHT AND EASY-TO-USE IMPLEMENTATION

All implementations in our framework and pipelines are built on PyTorch and Huggingface Transformers (Wolf et al., 2020). For defense evaluation, we employ the Hugging Face Trainer API to fine-tune the models. All resulting classification checkpoints will be publicly available on the Hugging Face Model Hub and can be easily downloaded and applied by researchers for further studies.

## 3.7 A LIGHTWEIGHT VISULIZATION FRAMEWORK

In our framework, we also introduce a visualization tool that enables users to explore how adversarial perturbations affect model predictions on input DNA sequences. Unlike language and computer vision domains—where explanations often rely on heuristic attribution or prediction maps—our approach leverages genomic knowledge to validate sequence-level changes with biological expectations. Although there is a growing body of literature on explainable AI in the context of adversarial attacks (Moshe et al., 2024; Devabhakthini et al., 2023; Gipiškis et al., 2023; Ozbulak et al., 2021), these works predominantly rely on saliency-based methods. In contrast, GFMs offer a promising path forward by grounding explanations in real-world biological data and leveraging bioinformatics for more interpretable and trustworthy insights.

## 4 EVALUATIONS OF THE CURRENT ATTACKS AND DEFENSES

In this section, we conduct a series of experiments to assess the impact of adversarial attacks and defenses on the safety of GFMs. We use DNABERT-2 (Zhou et al., 2024), HyenaDNA (Nguyen et al., 2024b), Nucleotide Transformer (NT) (Dalla-Torre et al., 2024), NT2, and GenomeOcean (Zhou et al., 2025b) as the target models.

**Models.** Following Zhou et al. (2024), we use DNABERT-2, NT, NT2, GenomeOcean, and HyenaDNA as target models which are trained specifically on DNA sequences. The first four are transformer-based models, whereas HyenaDNA utilizes a Hyena-based architecture. We finetune all models using the sequence classification technique, following Zhou et al. (2024), and utilize the finetuned models as the targets to evaluate the adversarial attacks—we generate adversarial examples that are misclassified by the target models while indistinguishable from the original examples.

**Datasets.** We utilize 26 datasets covering 5 tasks and 4 species, as detailed in Zhou et al. (2024). These datasets are specifically curated for genome sequence classification tasks, featuring input sequence lengths that range from 70 to 1000.

**Evaluation metrics.** We evaluate the effectiveness of adversarial attacks using the Attack Success Rate (ASR) and assess defense strategies using the Defense Success Rate (DSR) as detailed in appendix K.2. Accuracy is used as the core metric to quantify the impact of both attacks and defenses.

### 4.1 EVALUATING ADVERSARIAL ATTACKS ON GFMs

We utilize the same datasets and models as described in section 3.2 to ensure consistency in our evaluation. We conduct each evaluation three times with different random seeds and present the average and standard deviation for each metric.

**Baseline attack artifacts.** We test four baseline attack methods—BertAttack (Li et al., 2020a), TextFooler (Jin et al., 2020), PGD (Madry et al., 2018b), and FIMBA (Skovorodnikov & Alkhzaimi, 2024)—to assess their effectiveness in generating adversarial examples. Experiments are conducted

| | | Transformer-based | | | | Hyena-based |
| | | DNABERT-2 | NT2 | NT | OG | HyenaDNA |
|---|---|---|---|---|---|---|
| Epigenetic Marks Prediction | H3 | 3 | 4 | 2 | 5 | 1 |
| | H3K4me1 | 4 | 2 | 3 | 5 | 1 |
| | H3K4me2 | 2 | 1 | 3 | 4 | 5 |
| | H3K4me3 | 4 | 2 | 3 | 5 | 1 |
| | H3K14ac | 5 | 2 | 4 | 3 | 1 |
| | H3K36me3 | 3 | 1 | 2 | 4 | 5 |
| Epigenetic Marks Prediction | H3K9ac | 4 | 5 | 2 | 3 | 1 |
| | H3K79me3 | 3 | 2 | 4 | 5 | 1 |
| | H4 | 3 | 2 | 5 | 4 | 1 |
| | H4ac | 5 | 3 | 2 | 4 | 1 |
| Promoter Detection | prom_300_all | 2 | 4 | 3 | 5 | 1 |
| | prom_300_notata | 1 | 2 | 4 | 3 | 5 |
| | prom_300_tata | 4 | 2 | 3 | 1 | 5 |
| | prom_core_all | 4 | 1 | 3 | 5 | 2 |
| | prom_core_notata | 2 | 4 | 5 | 3 | 1 |
| | prom_core_tata | 2 | 1 | 4 | 3 | 5 |
| Transcription Factor Prediction (Hunan) | tf0 | 2 | 4 | 3 | 1 | 5 |
| | tf1 | 2 | 4 | 3 | 1 | 5 |
| | tf2 | 4 | 2 | 1 | 3 | 5 |
| | tf3 | 1 | 3 | 2 | 4 | 5 |
| | tf4 | 2 | 4 | 3 | 1 | 5 |
| Transcription Factor Prediction (Mouse) | mouse_0 | 4 | 5 | 3 | 2 | 1 |
| | mouse_1 | 1 | 4 | 5 | 3 | 2 |
| | mouse_2 | 4 | 2 | 5 | 3 | 1 |
| | mouse_3 | 2 | 3 | 1 | 4 | 5 |
| | mouse_4 | 3 | 2 | 1 | 4 | 5 |

Figure 3: **Performance of Adversarial Attacks on Different Model Architectures.** We assess the effectiveness of the evaluated adversarial attacks across diverse model architectures, including both transformer-based models (DNABERT-2, NT, NT2, GenomeOcean) and Hyena-based model (HyenaDNA). We use the Attack Success Rate (ASR) as the primary metric to evaluate the performance of the evaluated adversarial attacks. For each experiment, we rank the top five models based on their ASR, with ranks assigned from 1 to 5. Rank 1 denotes the most robust model (lowest ASR) and rank 5 denotes the least robust (highest ASR). Our results highlight how each model performs under attack, revealing differences in vulnerability and resilience across the architectures.

Table 1: **Adversarial Attack Performance of the Evaluated Method.** We conduct experiments to assess the effectiveness of the evaluated attack method against target models. The table presents a comparison of target model performance before and after applying the evaluated attack. We report Attack Success Rate (ASR) as the primary evaluation metric, with variance omitted as they are all $\leq 2\%$. The final columns present the average Attack Success Rate (ASR) across all GFM models for each specific attack. The last row similarly shows the average ASR across all attacks for each specific GFM. Additionally, for each attack, individual ASR scores are ranked from highest to lowest, with the rank displayed in brackets next to the score.

| | Transformer-based | | | | Hyena-based | |
| Attack | DNABERT-2 | NT | NT2 | GenomeOcean | HyenaDNA | Avg |
|---|---|---|---|---|---|---|
| BertAttack | 96.23%(5) | 99.87%(1) | 99.56%(4) | 99.57%(3) | 99.75%(2) | 99.00% |
| TextFooler | 92.37%(4) | 96.69%(2) | 96.56%(3) | 99.54%(1) | 88.45%(5) | 94.72% |
| PGD | 38.28%(2) | 38.23%(3) | 34.41%(5) | 36.57%(4) | 47.94%(1) | 39.09% |
| FIMBA | 39.94%(2) | 37.66%(3) | 36.50%(4) | 41.06%(1) | 30.35%(5) | 37.10% |
| Attack ASR | 66.71% (3.25) | 68.11% (2.25) | 66.76% (4) | 69.19% (2.25) | 66.62% (3.25) | |

on 5 GFMs, covering both transformer-based and Hyena-based architectures, with implementation details provided in appendix K.3. Attack performance is primarily measured using ASR, and methods are ranked based on their average ASR across all datasets.

**Results.** In Figure 3 and Table 1, our results highlight the effectiveness of the evaluated attacks in generating adversarial examples that are misclassified by target models. Further results are included in Table 7. We have below observations.

- GenomeOcean Zhou et al. (2025b) exhibits greater susceptibility to adversarial attacks than classification models (DNABERT-2, NT2), as evidenced by higher ASR and ranks across all GFMs. This observation aligns with the findings in Ebrahimi et al. (2018); Wang et al. (2023).

- NT2 demonstrates the highest robustness, indicated by its lowest average rank, potentially due to its use of BPE tokenization. GFMs employing BPE tokenization (DNABERT-2, NT2) appear to be more robust than those using k-mer tokenization (NT). BPE's subword structure allows for partial token retention despite alterations, hindering significant semantic or biological shifts. Interestingly, while NT2's average ASR is higher than HyenaDNA's (the lowest overall), its ASR rank is lower. In contrast, NT shares the highest ASR rank with GenomeOcean but has a lower ASR. The discrepancy stems from NT consistently achieving high ASR across all attacks, while GenomeOcean performs best on TextFooler and FIMBA but poorly on BertAttack and PGD.

- BertAttack yields the highest average ASR across GFMs, while FIMBA, the only genome-specific attack, shows the lowest, indicating limited effectiveness. This ineffectiveness may be due to constraints in the released FIMBA code [1] and evaluation setup in Skovorodnikov & Alkhzaimi (2024). However, traditional NLP-based adversarial attacks such as BertAttack and TextFooler already achieve a high ASR in these models. This underscores the importance of developing defense mechanisms tailored for GFM tasks to ensure their safety.

## 4.2 EVALUATING ADVERSARIAL DEFENSES

Each experiment is repeated three times with different random seeds on the same datasets and models, and we report the mean and standard deviation of each evaluation metric.

**Baseline defenses.** We assess the robustness of five GFM models against adversarial attacks using three defense baselines: adversarial training (Zheng et al., 2020) (employing TextFooler for data augmentation), FreeLB (Zhu et al., 2020), and ADFAR (Bao et al., 2021). Defenses were evaluated against BertAttack, TextFooler, and PGD attacks, with the DSR as the primary robustness metric. Further results are included in Table 8.

**Results.** As shown in Table 2, we have below observations:

- ADFAR achieves the highest overall DSR, significantly outperforming other defenses against BertAttack and TextFooler. However, ADFAR performs poorly against the PGD attack.

- FreeLB obtains better DSR against PGD, possibly due to it smooths the adversarial loss during training, which somewhat improves robustness.

- AT is less effective than ADFAR and FreeLB against BertAttack and TextFooler, although AT performs comparably to FreeLB against PGD attacks.

- While the model architecture does not significantly affect overall defense performance, specific models show distinct advantages, e.g., DNABERT-2 and NT2 show a greater defense improvement against BertAttack, while HyenaDNA demonstrates a better defense against TextFooler and PGD.

## 4.3 VISUALIZATION OF ADVERSARIAL ATTACKS

In this experiment, we visualize adversarial attacks on target models with our framework. We utilize BertAttack to generate adversarial examples and visualize the results using the DNABERT-2 model. The visualization highlights the subsequences that are most significant for the model's classification performance, specifically focusing on the frequency with which the adversarial attack modifies the sequence. We present the frequency of subsequence changes at the subword tokenizer level using Byte Pair Encoding (BPE). As shown in Figure 4, the visualization is generated by analyzing the frequency of subsequence changes across all datasets and models, providing insight into the most critical subsequences for the model's classification performance.

## 4.4 PERFORMANCE OF MODEL AUGMENTED WITH GENOADV DATASET

In order to show the effectiveness of the GenoAdv dataset, we conduct experiments to evaluate the performance of the model augmented with the GenoAdv dataset. We use BertAttack, TextFooler, and

---

[1]https://github.com/HeorhiiS/fimba-attack

Table 2: **Defense Performance Under Adversarial Attacks.** We conducted experiments to evaluate the performance of a defense method against adversarial attacks. The table compares the performance of target models, both with and without the evaluated defense, under BertAttack, TextFooler, and PGD attacks. The Defense Success Rate (DSR) is used as the primary evaluation metric, with variance omitted as they are all $\leq 2\%$. The best DSR values are highlighted in bold. In the table, **AT** denotes traditional adversarial training. We observe that ADFAR is the most effective defense based on DSR, particularly against BertAttack and TextFooler.

| Attack Method | Defense | Transformer-based | | | | Hyena-based |
| | | DNABERT-2 | NT | NT2 | GenomeOcean | HyenaDNA |
|---|---|---|---|---|---|---|
| BertAttack | N/A | 3.77% | 0.13% | 0.44% | 0.43% | 0.25% |
| | AT | 4.06% | 0.21% | 0.46% | 0.60% | 0.81% |
| | FreeLB | 4.34% | 0.67% | 0.71% | **2.94%** | 1.12% |
| | ADFAR | **21.84%** | **4.95%** | **6.96%** | 1.18% | **1.50%** |
| PGD | N/A | 61.73% | 61.77% | 65.59% | 63.43% | 52.06% |
| | AT | **64.92%** | 79.10% | 82.02% | **66.14%** | 85.67% |
| | FreeLB | 64.07% | **79.38%** | **88.53%** | 65.96% | **86.99%** |
| | ADFAR | 63.48% | 63.44% | 72.89% | 65.87% | 83.74% |
| TextFooler | N/A | 7.63% | 3.31% | 3.44% | 0.46% | 11.55% |
| | AT | 20.97% | 42.88% | 18.95% | 18.51% | **84.19%** |
| | FreeLB | 18.39% | 42.94% | 18.16% | 17.33% | 69.56% |
| | ADFAR | **32.88%** | **67.07%** | **22.00%** | **46.18%** | 80.82% |

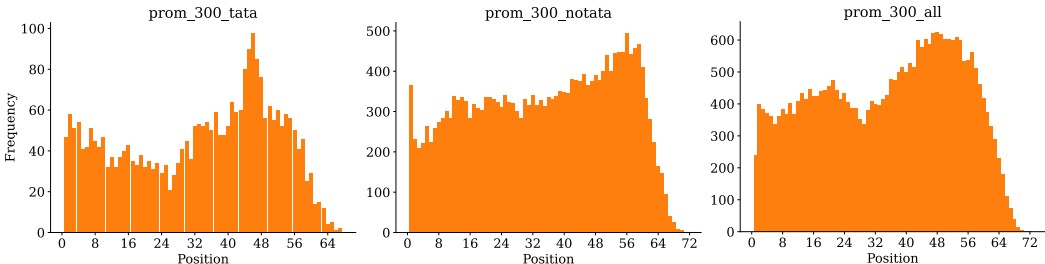

Figure 4: **Examples of the visualization of GFMs with adversarial attacks.** We present the results of the three tasks of the DNABERT-2 model under BertAttack. All subsequence changes occur at the subword tokenizer level using Byte Pair Encoding (BPE) (Sennrich et al., 2016). The visualization highlights which parts of the sequence are most significant for the model's classification performance. Specifically, we present the frequency with which the adversarial attack modifies the sequence. A higher frequency suggests that the subsequence plays a more important role in the model's classification decisions.

PGD to evaluate the DSR on 5 GFMs. In our experiment, we perform traditional adversarial training with TextFooler-augmented data as a baseline, and compare it to the same training approach using the GenoAdv dataset. We conduct each evaluation three times with different random seeds and present the average and standard deviation for each metric.

**Results:** As shown in Table 3, adversarial training with GenoAdv data yields stronger robustness against adversarial attacks compared to training with only TextFooler-augmented samples in most cases. This suggests that the GenoAdv dataset offers valuable augmentation data to mitigate the vulnerability of GFMs. Specifically, using GenoAdv data to do data augmentation leads to a performance improvement of 34.71% over TextFooler. Further results are included in Table 9.

## 4.5 QUANTIZATION INFLUENCE ON ADVERSARIAL ATTACKS

To evaluate the influence of quantization on evaluated attacks, we conduct experiments on quantized versions of target models. Inside those quantization methods, some of them are based on the traditional quantization methods, such as uniform quantization, and some of them are based on the outluer-removal quantization methods, such as OutEffHop (Hu et al., 2024). Following the quantization setup in Luo et al. (2025) and Wu et al. (2025), we evaluate the performance of the attacks on quantized models with 8-bit weights and 8-bit activations (W8A8), comparing them to the original models to analyze the impact of quantization on attack detectability.

Table 3: **Defense Performance Augmented with the GenoAdv Dataset.** We conduct experiments to evaluate the performance of a model augmented with the GenoAdv dataset against adversarial attacks. The table compares the performance of the target models, both with and without the GenoAdv dataset augmentation, under BertAttack, TextFooler, and PGD attacks. We report ASR as the primary evaluation metric, with variance omitted as they are all $\leq 2\%$. The best results are highlighted in bold. In the table, **AT** denotes traditional adversarial training. We observe that GenoAdv samples are more effective than TextFooler samples under traditional adversarial training methods.

| Attack Method | Defense | Transformer-based | | | | Hyena-based |
|---|---|---|---|---|---|---|
| | | DNABERT-2 | NT | NT2 | GenomeOcean | HyenaDNA |
| BertAttack | N/A | 3.77% | 0.13% | 0.44% | 0.43% | 0.25% |
| | AT | 4.06% | 0.21% | 0.46% | 0.60% | 0.81% |
| | GenoAdv | **5.17%** | **0.69%** | **0.59%** | **0.73%** | **5.23%** |
| PGD | N/A | 61.73% | 61.77% | 65.59% | 63.43% | 52.06% |
| | AT | 64.92% | 79.10% | **82.02%** | 66.14% | **85.67%** |
| | GenoAdv | **69.32%** | **79.31%** | 75.57% | **67.10%** | 84.52% |
| TextFooler | N/A | 7.63% | 3.31% | 3.44% | 0.46% | 11.55% |
| | AT | 20.97% | 42.88% | 18.95% | 18.51% | **84.19%** |
| | GenoAdv | **22.19%** | **44.05%** | **20.56%** | **19.45%** | 81.99% |

Table 4: **Performance of the evaluated attacks on quantized models.** We perform experiments to assess how quantization affects the effectiveness of adversarial attacks on target models. The table compares model performance before and after quantization under BertAttack and TextFooler attacks. Attack Success Rate (ASR) serves as the primary evaluation metric, with variance omitted as they are all $\leq 2\%$. The best results are highlighted in bold.

| Attack Method | Model | Quantized Method | ASR ($\downarrow$) |
|---|---|---|---|
| BertAttack | DNABERT-2 | - | 96.23 |
| | | Vanilla | **59.46** |
| | | OutEffHop | 64.71 |
| | NT1 | - | 99.87 |
| | | Vanilla | **99.37** |
| | | OutEffHop | 99.42 |
| TextFooler | DNABERT-2 | - | 92.37 |
| | | Vanilla | **19.90** |
| | | OutEffHop | 21.34 |
| | NT1 | - | 98.23 |
| | | Vanilla | **66.57** |
| | | OutEffHop | 68.53 |

**Results.** In Table 4, our results highlight the effectiveness of quantization in improving the robustness of target models against adversarial attacks. Specifically, we observe that the evaluated attacks achieve a lower ASR on quantized models compared to the original models, indicating that quantization strengthens the defenses against these attacks. Additionally, the outlier-free quantization method also reduces the ASR of the evaluated attacks. This outcome suggests that quantization can improve model robustness against adversarial attacks. One possible explanation is that quantization introduces "flat regions" in the loss landscape, which diminishes the model's sensitivity to small perturbations. This observation aligns with the findings reported in Lin et al. (2019).

However, we find that the OutEffHop quantization method results in a higher ASR compared to traditional quantization methods, indicating that outlier-removal quantization can compromise the robustness of target models against adversarial attacks. A possible reason is that OutEffHop removes attention outliers which improves quantization, but also eliminates "flat regions" in the loss landscape that are important for robustness in traditional quantization methods. We also find that quantization significantly impacts DNABERT-2 models, but has minimal effect on NT1 models, suggesting model-specific robustness gains. Notably, TextFooler is more affected by quantization than BERT-Attack, likely due to its dependence on precise word importance scores and synonym substitutions, which are disrupted by quantization-induced shifts in decision boundaries.

## 5 ANALYSIS AND INSIGHTS

In this section, we synthesize the experimental findings from section 4 to highlight distinctive observations regarding the adversarial robustness of Genomic Foundation Models (GFMs), contrasting them with established paradigms in Natural Language Processing (NLP) and Computer Vision (CV). Based on these insights, we outline critical directions for advancing attack and defense strategies in genomics.

(1) **Hypersensitivity to Minimal Perturbations:** Unlike natural language, genomic sequences lack redundancy which means minimal nucleotide-level changes are sufficient to flip predictions, highlighting the high semantic density of genomic tokens compared to linguistic subwords.

(2) **Tokenization and Robustness:** Tokenization granularity impacts vulnerability. On the Epigenetic Marks Prediction task, single-nucleotide models like HyenaDNA demonstrate superior robustness compared to multi-nucleotide aggregation methods, including BPE-based and $k$-mer-based models. This suggests that coarser tokenization amplifies sensitivity to local perturbations.

(3) **Semantic Alignment Vulnerability:** As visualized in Figures 4 and 6, adversarial attacks do not occur randomly but disproportionately target biologically functional motifs. This "semantic alignment vulnerability" indicates that while models correctly attend to critical regions, they remain brittle to imperceptible manipulations within them.

(4) **Quantization as Defense:** As evidenced in Table 4, quantization consistently lowers the Attack Success Rate (ASR) by flattening the loss landscape. However, outlier-free quantization offers slightly less protection than standard methods, suggesting that attention outliers may paradoxically contribute to stability against perturbations.

(5) **Limitations of Standard Adversarial Training:** Standard Adversarial Training underperforms because it relies on discrete, heuristic perturbations used by TextFooler. As shown in Table 2, this generates a narrow training manifold that fails to cover gradient-driven failure modes, leading to overfitting on specific edit patterns rather than achieving generalized robustness against attacks like BertAttack or PGD.

(6) **Mechanism of Effective Defenses:** Robustness in ADFAR and FreeLB stems from landscape optimization rather than data augmentation. Our ablation study in Table 12 reveals that random mutation yields negligible gains of $\sim 4\%$ DSR, compared to ADFAR with $\sim 22\%$ DSR. ADFAR succeeds by penalizing over-reliance on high-frequency motifs, effectively flattening the local loss in biologically sensitive regions, while FreeLB improves robustness by smoothing the adversarial loss in the embedding space.

## 5.1 POTENTIAL DIRECTIONS FOR ADVANCING ADVERSARIAL ATTACK AND DEFENSE

Based on the above observations, we propose the following potential attack and defense strategies:

- **Defense.** Our observations on the effectiveness of ADFAR and the model's high sensitivity to motif substitution highlight a promising defense direction centered on motif variability. Models often memorize specific motif instances (e.g., TATAAT), making them vulnerable to simple substitutions. Incorporating biological synonym augmentation during training can mitigate this issue. By sampling motif variants from databases such as JASPAR PWMs and including functionally equivalent alternatives (e.g., TATAAA), models can learn the probabilistic structure of motifs rather than overfitting to a single canonical sequence. This variability can further complement existing robustness mechanisms such as single-nucleotide tokenization and loss-smoothing regularization.

- **Attack.** The significant robustness gap between k-mer and single-nucleotide models suggests the need for a more genomics-aware attack strategy. A tokenizer-adaptive approach which first identifies the target model's tokenization scheme and then applies the most effective attack for that scheme can more efficiently exploit vulnerabilities. Substitution-based attacks are particularly harmful for k-mer models, whereas gradient-based perturbations are more effective for single-nucleotide models. Aligning these complementary strategies may yield stronger and more targeted adversarial methods.

## 6 DISCUSSION AND CONCLUSION

We introduce GenoArmory, the first unified adversarial attack benchmark for DNA-based Genomic Foundation Models (GFMs). Our benchmark offers an accessible, reproducible, and comprehensive framework, enabling users to confidently evaluate and compare adversarial robustness in GFMs. Also, to encourage broad participation, we do not restrict the architectures of target models. Instead, GenoArmory offers a standardised framework for evaluating adversarial attacks and defenses. Methodologically, compared to adversarial attack benchmarks in language and computer vision (Zheng et al., 2023a; Croce et al., 2021; Dong et al., 2020), GenoArmory includes visualization tools that facilitate deeper insights into the evaluated attacks—leveraging the fact that GFM data is inherently structured and scientifically meaningful.

## ETHICS STATEMENT

This work investigates adversarial vulnerabilities of genomic foundation models (GFMs) and proposes defensive strategies to improve their robustness. In line with the ICLR Code of Ethics[2], we explicitly acknowledge the dual-use nature of our contributions and have taken a series of measures to mitigate potential misuse in appendix E. While adversarial attack methods and the **GenoAdv** dataset may expose weaknesses in existing GFMs, our intention is to promote transparency, reproducibility, and proactive defense development.

To proactively address these risks and uphold our commitment to responsible research, we have implemented several key measures. First, we have disclosed our findings to all GFM development teams in our paper and received several responses prior to publication, more details are available in appendix J. Second, we restricted the released code and dataset to non-commercial research purposes only, explicitly discouraging their use in clinical or diagnostic applications. Data generated directly from adversarial attacks will not be publicly available and securely stored. All access requests must be directed to the authors and will undergo evaluation by the team's healthcare cybersecurity professionals. Further details are provided in appendix E.1. We emphasize that the adversarial artefacts provided are intended solely to facilitate research on model robustness, biosecurity, and responsible AI in genomics.

No human subjects or personally identifiable genomic data are involved in this study. All datasets are synthetic or derived from publicly available non-sensitive genomic resources. We are committed to ensuring that our work advances scientific understanding while safeguarding public health and safety.

## REPRODUCIBILITY STATEMENT

To ensure reproducibility, we release an anonymous open-source code repository containing the full implementation of the GenoArmory framework, including pipelines for adversarial attacks, defenses, and evaluation, as well as the GenoAdv dataset and adversarial artifact repository, which is provided in Appendix A. All experiments are run with three random seeds, and results remain highly stable with reported standard deviations below 2%. We employ a unified training setup using the AdamW optimizer with learning rate $3 \times 10^{-5}$, batch size 64, maximum sequence length 256, warmup ratio 0.05, and training for up to 4 epochs. Representative hyperparameter configurations are reported in the appendix (e.g., BertAttack with $k = 48$, ADFAR with frequency threshold $f_{\text{thres}} = 200$, FreeLB with adversarial magnitude 0.6), with full details available in Appendix F. We will release checkpoints for all evaluated Genomic Foundation Models to support independent verification and extension.

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

# Supplement Material

## A OPEN SCIENCE

We plan to release the code, pretrained checkpoints, and datasets used in our work upon acceptance. The code will be made available at our GitHub repository, and the pretrained checkpoints will be hosted on HuggingFace. The GenoAdv dataset will also be shared on the HuggingFace Datasets platform. For now, codes are available at: https://anonymous.4open.science/r/GenoArmory-78C0.

## B  FUTURE WORK

Although GenoArmory provides a comprehensive evaluation of adversarial attacks and defenses on DNA-based GFMs, it still has several limitations. For example, GenoArmory currently excludes RNA-based GFMs and is limited to classification tasks, leaving other task types and modalities unaddressed.

**Developing a comprehensive benchmark** is essential, as GFM safety is often underestimated. Yet, insufficient safeguards hinder their advancement and pose risks to scientific progress. A key challenge in improving GFM safety is the lack of a comprehensive benchmark for evaluating vulnerabilities. In this paper, we provide the **first** in-depth analysis of DNA-based attacks on leading GFMs using such a benchmark. However, this serves only as a foundation—future work must extend it to include broader attack vectors, such as RNA-based model attacks, to ensure more robust evaluation. Greater focus is also needed on generative GFMs, such as Evo (Nguyen et al., 2024a), which remain underrepresented in safety evaluations. Beyond benchmarks, the lack of automated tools for assessing the safety of generated genomic sequences—unlike in image or speech domains—poses a critical gap. This highlights the urgent need for robust, domain-specific evaluation frameworks to ensure safe and ethical deployment of GFMs.

**Automatic sequence data judgment system** provides a framework for assessing sequence differences to evaluate the safety of generated genomic sequences. Prior work on sequence functionality (Sim et al., 2012; Flanagan et al., 2010) and ortholog analysis (Jensen, 2001) demonstrates that ortholog comparisons can reveal relationships between genomic sequences, informing safety assessments. Building on this idea, Emms & Kelly (2019) introduce a method to calculate ortholog differences within genomic sequences. By using the distance between sequence orthologs, researchers can quantify differences between generated sequences and known harmful genomic sequences, providing a method to assess sequence safety. This approach enables the development of an automated system for sequence evaluation, improving efficiency in safety assessments. Additionally, leveraging large language models (LLMs) like Qwen (Chu et al., 2023) and Llama3 (Dubey et al., 2024) to generate genomic sequences enhances the model's diversity and robustness.

## C  BOARDER IMPACT

This paper seeks to advance the trustworthiness of genomic foundation models (GFMs). While the work does not have immediate social implications, it represents a step toward creating more reliable GFMs. However, the adversarial samples released in the **GenoAdv** dataset and experiments can provide incorrect classification for existing GFMs.

## D  RELATED WORK

In this section, we explore the background of vulnerabilities in GFMs. We begin by introducing benchmarks for evaluating adversarial attacks on GFMs, including standard datasets, metrics, and evaluation protocols. Next, we review existing adversarial attack methods tailored for GFMs, such as BERT-Attack (Li et al., 2020a) and PGD (Madry et al., 2018b). Finally, we discuss defense strategies against these attacks, covering approaches like FreeLB (Zhu et al., 2020) and ADFAR (Bao et al., 2021).

### D.1  BENCHMARKS

The GUE benchmark (Zhou et al., 2024) encompasses a variety of genome classification tasks, including promoter detection, transcription factor prediction, and COVID variant classification. These tasks are designed to assess model performance across multiple species, such as humans, fungi, viruses, and yeast. Building on this, GUE+ extends the benchmark to focus on tasks involving longer input sequences, ranging from 5000 to 10000 base pairs, to evaluate models' capabilities in processing and analyzing complex genomic data. The GUE benchmark assesses model performance using metrics such as Accuracy, F1-score, and Matthews Correlation Coefficient (MCC) (Chicco & Jurman, 2020).

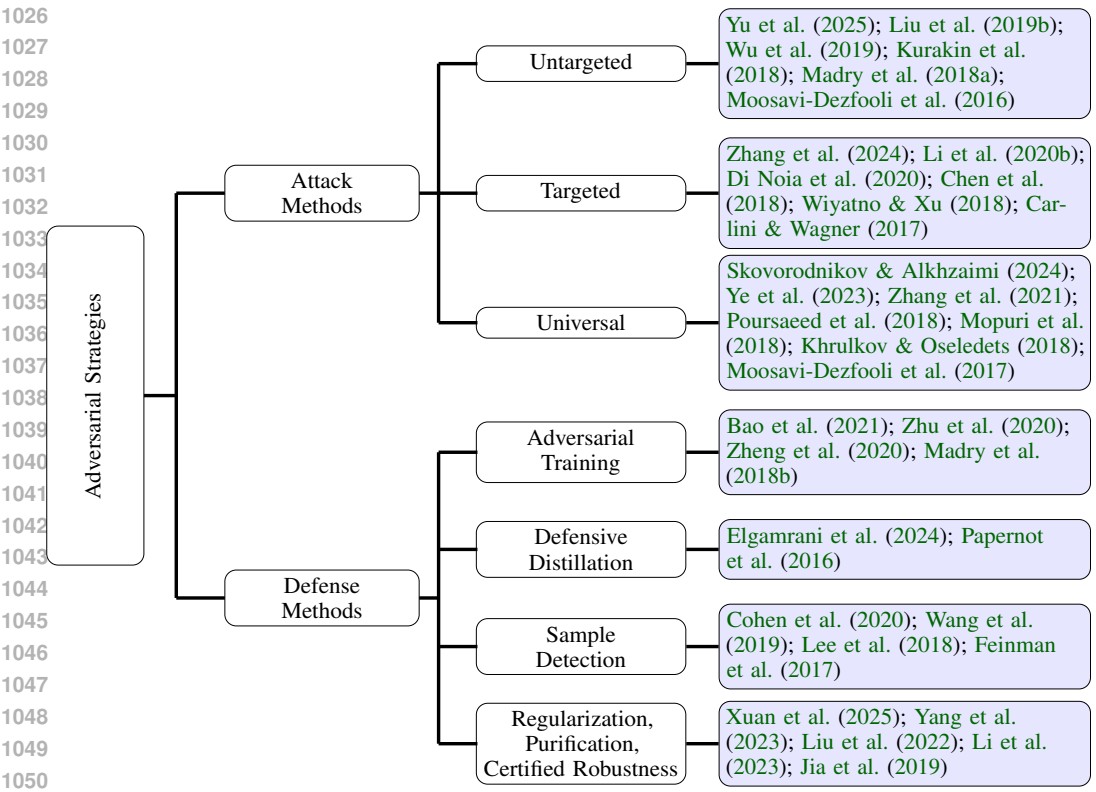

Figure 5: **Taxonomy of Adversarial Strategies.**

Meanwhile, GenBench (Liu et al., 2025) is a comprehensive benchmarking suite tailored for evaluating the performance of GFMs. It systematically analyzes datasets from diverse biological domains, with a focus on both short-range and long-range genomic tasks. These tasks encompass essential areas such as coding regions, non-coding regions, and genome structure. For classification tasks, GenBench uses cross-entropy loss to measure prediction divergence and evaluates performance with top-1 accuracy and AUC-ROC. For regression tasks, it applies Mean Squared Error (MSE) for accuracy and calculates Spearman and Pearson correlation coefficients to assess relationships.

These benchmarks (Liu et al., 2025; Grešová et al., 2023) offer a thorough evaluation of GFMs. However, all these benchmarks overlook the safety aspects of the GFMs. Recently, the safety of large scientific foundation models has become a prominent focus in research (Li et al., 2024; Skovorodnikov & Alkhzaimi, 2024). As a groundbreaking approach to incorporating adversarial attacks into genomic data analysis, FIMBA (Skovorodnikov & Alkhzaimi, 2024) leverages publicly available genomic datasets, such as The Cancer Genome Atlas (TCGA) and COVID-19 single-cell RNA sequencing data, to assess the robustness of AI models against adversarial feature importance attacks. In the TCGA dataset, the classification task aims to determine whether a sample is malignant, while in the COVID-19 dataset, the objective is to identify whether a patient is diagnosed with the disease. As part of this evaluation, FIMBA uses Accuracy as the primary performance metric to measure the classification capability. To assess the quality and stealth of the adversarial attacks, they employ the Structural Similarity Index Measure (SSIM). SSIM quantifies the structural similarity between the original and adversarially attacked data, with higher values indicating attacks that are more undetectable and preserve the data's original structure.

### D.2 ADVERSARIAL ATTACK

Adversarial attacks can be broadly classified into untargeted, targeted, and universal attacks. Untargeted attacks (Yu et al., 2025; Liu et al., 2019b; Wu et al., 2019; Kurakin et al., 2018; Madry et al., 2018a; Moosavi-Dezfooli et al., 2016) aim to cause any misprediction by modifying the input in the direction of the loss gradient, maximizing overall loss. In contrast, targeted attacks (Zhang

et al., 2024; Li et al., 2020b; Di Noia et al., 2020; Chen et al., 2018; Wiyatno & Xu, 2018; Carlini & Wagner, 2017) guide the model's output toward a specific attacker-defined class using the loss gradient directed at the target class. Universal attacks (Skovorodnikov & Alkhzaimi, 2024; Ye et al., 2023; Zhang et al., 2021; Poursaeed et al., 2018; Mopuri et al., 2018; Khrulkov & Oseledets, 2018; Moosavi-Dezfooli et al., 2017) generate perturbations applicable to any input from a given class, causing mispredictions universally.

The Fast Gradient Sign Method (FGSM) (Liu et al., 2019b) and Projected Gradient Descent (PGD) (Madry et al., 2018b) are two prominent techniques for generating adversarial examples in machine learning, particularly for deep neural networks (Shayegani et al., 2023). FGSM generates adversarial samples by applying a single-step perturbation in the direction of the gradient of the loss function, scaled to a predefined magnitude, making it computationally efficient. However, PGD improves robustness by iteratively applying small gradient-based perturbations while ensuring that adversarial examples remain within a specified norm constraint, leading to more effective attacks.

A variety of adversarial attack and defense strategies have recently been proposed, specifically tailored for natural language processing (NLP) tasks (Goyal et al., 2023). These techniques can be categorized into character-level, word-level, and sentence-level adversarial attacks. Character-level adversarial attacks involve perturbing individual characters in text to mislead machine learning models while preserving readability. For example, DeepWordBug (Gao et al., 2018) modifies specific characters based on importance scores to maximize the model's misclassification while minimizing changes to the text. Similarly, TextBugger (Li et al., 2019) generates adversarial examples by replacing, inserting, or removing characters, focusing on semantic preservation and evading detection by defense mechanisms. Word-level adversarial attacks focus on perturbing entire words rather than individual characters. These attacks can be broadly classified into three categories: gradient-based, importance-based, and replacement-based methods. Gradient-based methods, such as FGSM (Liu et al., 2019b), utilize gradients to identify vulnerable words and modify them to maximize the model's loss. Importance-based methods, exemplified by TextFooler (Jin et al., 2020), rank words based on their contribution to the model's prediction and replace them with semantically similar alternatives to alter the output. Replacement-based methods, like BERT-Attack (Li et al., 2020a), leverage pre-trained language models to generate context-aware substitutions, ensuring the adversarial examples maintain fluency and semantic coherence. Sentence-level adversarial attacks involve generating adversarial examples by modifying entire sentences to mislead the model while maintaining grammaticality and semantic relevance. AdvGen (Cheng et al., 2019) generates adversarial sentences by leveraging reinforcement learning to iteratively modify sentence structures and word choices, ensuring the adversarial examples remain coherent and natural while effectively deceiving the target model.

Adversarial attacks have also been explored in genomic models to assess their robustness and identify vulnerabilities in sequence-based predictions. FIMBA (Skovorodnikov & Alkhzaimi, 2024) presents a black-box, model-agnostic attack and analysis framework designed for widely used machine learning models in genomics. FIMBA targets genomic models by perturbing key features identified through SHAP values, which measure the importance of each feature to the model's decision. By selecting the most impactful features and modifying them using interpolation between the original and target vectors, FIMBA generates minimally altered adversarial examples that effectively deceive the model. The attack avoids gradient reliance, functioning as a black-box method, and focuses on modifying as few features as possible to ensure both high efficacy and low detectability.

### D.3 ATTACK METHOD CATEGORIZATION

To provide a comprehensive evaluation, we categorize our adversarial attack methods as follows:

- **White-box attacks:**
    - PGD
    - BertAttack and TextFooler
    - AutoAttack (ensemble of gradient-based methods)
    - UAP
- **Black-box attacks:**
    - FIMBA

- **Targeted vs. Untargeted settings:**
  All attacks except PGD and AutoAttack are performed in the untargeted setting. PGD and AutoAttack are evaluated in both targeted and untargeted modes.

## D.4 Defense Methods

To improve the robustness of GFMs, various defense strategies (Ke et al., 2025; Luo et al., 2024; Bao et al., 2021; Zhu et al., 2020; Cohen et al., 2020; Lee et al., 2018; Papernot et al., 2016) are proposed, including adversarial training, defensive distillation, adversarial sample detection, and regularization, purification, and certified robustness. Among these, adversarial training (Bao et al., 2021; Zhu et al., 2020; Zheng et al., 2020; Madry et al., 2018b) is the most effective, enhancing model resilience by injecting adversarial examples during training. Among these methods, Madry et al. (2018a) propose a method to inject bounded perturbations into word embeddings and minimize worst-case loss, almost halving BERT-Attack and TextFooler success rates without degrading clean accuracy. FreeLB (Zhu et al., 2020) merges several PGD steps into one forward-backward pass and accumulates gradients, cutting training cost; FreeLB++ (Li et al., 2021) enlarges the radius and steps for further robustness gains at no extra accuracy loss. Other lightweight variants such as SMART(Jiang et al., 2020), TAVAT (Li & Qiu, 2021), and R3F (Aghajanyan et al., 2020) approximate the inner maximization with uncertainty- or noise-based regularization, reaching performance close to FreeLB++ at a fraction of the compute. The frequency-aware randomization framework ADFAR (Bao et al., 2021) incorporates anomaly-detection signals and word-frequency constraints directly into the training loop, unifying adversarial sample detection ideas with adversarial training to further weaken substitution-based attacks without extra overhead. Defensive distillation (Elgamrani et al., 2024; Papernot et al., 2016) trains a student model on softened outputs from a teacher model to smooth decision boundaries, though its efficacy against strong adversarial attacks remains debated. However, Carlini & Wagner (2016) demonstrate that defensive distillation is ineffective against adaptive adversarial attacks, as carefully crafted inputs can still bypass the smoothed decision boundaries and fool the model. Adversarial sample detection (Cohen et al., 2020; Wang et al., 2019; Lee et al., 2018; Feinman et al., 2017) focuses on identifying malicious inputs rather than improving model robustness. MAFD (Jin et al., 2024) combines perplexity, word frequency, and masking-probability features for robust anomaly scoring; ONION (Qi et al., 2021) leverages language-model perplexity to prune high-risk tokens; Sharpness-based detectors (Zheng et al., 2023b) add infinitesimal noise and flag samples exhibiting steep loss increases. Deployed alongside adversarial training, these detectors offer real-time protection against unseen or cross-domain attacks. Regularization, purification and certified Robustness reduce perturbation sensitivity by modifying the loss or sanitizing inputs. Flooding-X (Liu et al., 2022) maintains a loss floor to guide the model toward flatter regions; adversarial label smoothing (Yang et al., 2023) and temperature scaling (Xuan et al., 2025) curb over-confidence; masked-language-model purification (Li et al., 2023) masks and reconstructs suspicious tokens to cleanse perturbations. Interval bound propagation (IBP) (Jia et al., 2019) and randomized smoothing schemes such as SAFER (Ye et al., 2020) and RanMASK (Zeng et al., 2023) provide formal guarantees against word substitutions or masking budgets.

# E Ethical Considerations

Prior to making this work public, we share our adversarial attack artefacts and our results with leading GFMs teams, as shown in appendix J. Secondly, we open-source the code and data used in our experiments to promote transparency. Also, we carefully consider the ethical impact of our work and list the two impacts: (1) The adversarial sample released in the **GenoAdv** dataset and experiments can provide incorrect classification for existing GFMs. (2) Adversarial training is an efficiency method to make GFMs more resilient to adversarial attacks.

## E.1 Dual-Use and Misuse Risks

We recognize that adversarial attacks on genomic foundation models (GFMs), particularly those applied to clinical diagnostics and gene pathogenicity prediction, raise significant dual-use and misuse concerns. While our intention is to improve the safety and robustness of GFMs, we acknowledge that, if misused, the techniques developed in this work could be repurposed to evade genomic screening, manipulate diagnostic predictions, or interfere with treatment decision-making.

The adversarial samples included in the **GenoAdv** dataset are designed to reveal vulnerabilities in current models by targeting biologically meaningful regions. These vulnerabilities highlight the urgency for robust defensive strategies. However, we also recognize that releasing such resources without caution could present opportunities for malicious use.

To mitigate these risks, we take the following steps. First, we have contacted several leading GFM development teams to disclose our findings and foster collaboration on model hardening. Second, although we open-sourced our code and data to promote reproducibility, we now include a usage statement specifying that the tools and dataset are intended strictly for non-commercial research purposes. All data directly generated from adversarial attacks will not be publicly released and will be securely stored; any organizations or individuals seeking access must directly contact the authors, and requests will be evaluated by healthcare cybersecurity professionals within our team. Use in clinical or diagnostic applications, or for purposes that could impact public health, is explicitly discouraged.

We urge future researchers to approach this line of work with similar responsibility. Any use of GenoAdv or our attack pipeline should be guided by ethical principles that prioritize model reliability, biosecurity, and societal benefit. Our overarching goal is not to facilitate harm, but to proactively identify and close security gaps in genomic models before they can be exploited in real-world settings.

# F    REPRODUCIBILITY

In this section, we provide a discussion on the reproducibility of our experiments, including the details of the datasets used, the training and evaluation protocols, and the hyperparameters employed in our experiments.

## F.1    SOURCE OF RANDOMNESS.

To ensure reproducibility, we run all experiments using three different random seeds. We observe that the results are highly stable, with the benchmark introducing only minor variations—showing a variance of at most 2%.

## F.2    IMPLEMENTATION.

To ensure reproducibility, we implement the adversarial attack and defense methods based on their official GitHub repositories, as shown below:

- **BertAttack:** https://github.com/LinyangLee/BERT-Attack
- **TextFooler:** https://github.com/jind11/TextFooler
- **PGD:** https://github.com/MadryLab/robustness
- **FIMBA:** https://github.com/HeorhiiS/fimba-attack
- **ADFAR:** https://github.com/LilyNLP/ADFAR
- **FreeLB:** https://github.com/zhuchen03/FreeLB
- **AutoAttack:** https://github.com/fra31/auto-attack
- **Universal:** https://github.com/LTS4/universal

## F.3    HYPERPARAMETER.

We present the hyperparameters used in the benchmark for each model. We use **AdamW** (Loshchilov, 2017) as the optimizer. Fine-tuning and adversarial training are performed uniformly across all models and datasets for 4 epochs, using a batch size of 64 and a maximum sequence length of 256. We use the AdamW optimizer with a learning rate of $3e^{-5}$, gradient accumulation steps of 1, and a warmup ratio of 0.05. The maximum sequence length and batch size used for each adversarial attack and defense method are summarized in Table 5. These settings are chosen to balance computational efficiency and attack effectiveness across different methods.

For **BertAttack**, we configure the attack with k = 48 and set the prediction score threshold to 0, using DNABERT-2 as the reference masked language model. In **ADFAR's** frequency-aware randomization

Table 5: Hyperparameter settings for each attack method.

| Hyperparameter | BertAttack | TextFooler | PGD | FIMBA | ADFAR | FreeLB |
|---|---|---|---|---|---|---|
| Max Sequence Length | 128 | 256 | 256 | 128 | 128 | 256 |
| Batch Size | 32 | 128 | 16 | 32 | 2 | 32 |

process, we set the frequency threshold $f_{\text{thres}} = 200$, the number of samples $n_s = 20$, and the number of features $n_f = 10$. For **FreeLB**, the hyperparameters used in our experiments include an adversarial learning rate of 0.1, adversarial magnitude of 0.6, two adversarial steps, a base learning rate of $1e^{-5}$, gradient accumulation steps set to 1, and a weight decay of $1e^{-2}$.

## F.4 ADAPTING GENOMEOCEAN FOR CLASSIFICATION TASKS

Although GenomeOcean was originally proposed as a generative Genomic Foundation Model, its architecture natively supports sequence classification through a task-specific fine-tuning head, as described in Zhou et al. (2025b) In our implementation, we directly employed this built-in classification interface by initializing the pretrained encoder weights and fine-tuning the classification head on each benchmark dataset. Specifically, the final encoder representation corresponding to the [CLS] token (or equivalent global embedding) was passed through a one-layer MLP. This setup preserves GenomeOcean's generative capacity while maintaining methodological consistency across all evaluated models.

## G ADVERSARIAL ATTACK DISTANCE DEFINITION

The distance metric $d(X, X')$ introduced in section 2 varies according to the attack method:

- **Substitution-based attacks (BertAttack, TextFooler, FIMBA):**

    - **Edit distance:** $d(X, X') = $ number of nucleotide positions where $X$ and $X'$ differ
    - **Constraint:** $d(X, X') \leq \varepsilon$, where $\varepsilon$ is the maximum allowed substitutions

- **Gradient-based attacks (PGD):**
  The distance is measured in the continuous embedding space using the $\ell_\infty$ norm:

$$\|\text{Embed}(X') - \text{Embed}(X)\|_\infty \leq \varepsilon.$$

**Biological interpretation:** The edit distance constraint reflects a limit on SNP-like mutations, ensuring that adversarial sequences remain within a biologically realistic mutational range.

## H DISCLOSURE OF LLM USAGE

We utilize Cursor to assist in writing repetitive bash automation scripts and employ GPT-4o to refine the paper's language for conciseness and precision.

## I ADDITIONAL GENOARMORY DEMONSTRATION

We provide two installation options for GenoArmory and two usage methods: via command line and Python code.

**Example of Installation of GenoArmory**

```
# Install with pip
pip install genoarmory

# Install with source code
git clone https://Anonymous.git
conda create -n genoarmory pip=3.9
pip install .
```

**Example of Python Usage of GenoArmory**

```
# Initialize model
from GenoArmory import GenoArmory
import json
# You need to initialize GenoArmory with a model and tokenizer.
gen = GenoArmory(model=None, tokenizer=None)
params_file = 'xxx/scripts/PGD/pgd_dnabert.json'

# Visulization
gen.visualization(
    folder_path='xxx/BERT-Attack/results/meta/test',
    output_pdf_path='xxx/BERT-Attack/results/meta/test'
)

# Attack
if params_file:
  try:
      with open(params_file, "r") as f:
          kwargs = json.load(f)
  except json.JSONDecodeError as e:
      raise ValueError(f"Invalid JSON in params file")
  except FileNotFoundError:
      raise FileNotFoundError(f"Params file not found.")

gen.attack(
    attack_method='pgd',
    model_path='Anonymous_Model',
    **kwargs
)
```

---

**Example of Commend Line Usage of GenoArmory**

```
# Attack
python GenoArmory.py
--model_path Anonymous_Model attack
--method pgd --params_file xxx/scripts/PGD/pgd_dnabert.json

# Defense
python GenoArmory.py
--model_path Anonymous_Model defense
--method at --params_file xxx/scripts/AT/at_pgd_dnabert.json

# Visualization
python GenoArmory.py
--model_path Anonymous_Model visualize
--folder_path xxx/BERT-Attack/results/meta/test
--save_path xxx/BERT-Attack/results/meta/test/frequency.pdf

# Read MetaData
python GenoArmory.py
--model_path Anonymous_Model read
--type attack --method TextFooler --model_name dnabert
```

---

**Example of Disclosure Letter**

Dear DNABERT/DNABERT-2/DNABERT-S team,
```
 We hope this message finds you well.  We are reaching out to
share the preliminary results and artifacts from our recent
study on adversarial attacks targeting DNA-based Genomic
Foundation Models (GFMs), which we plan to release publicly
as part of a unified benchmarking framework.
Given your leading role in the development of GFMs, we
believe it is essential to disclose our findings to you in
advance.  Our results demonstrate that carefully crafted
adversarial sequences can induce incorrect classifications
across multiple GFM architectures.  We also find that
adversarial training remains a promising defense strategy
for enhancing model robustness.
To support responsible disclosure, we are providing:
1.  A summary of key findings and model vulnerabilities
2.  The adversarial sample set and evaluation scripts
3.  A description of our ethical considerations and intended
safeguards
We welcome your feedback on potential risks, mitigation
strategies, and collaborative opportunities to ensure this
research contributes constructively to the GFM community.
Please let us know if you would like early access to the
materials or would prefer to schedule a meeting to discuss
further.
```
Best regards,
GenoArmory Author

## J DISCLOSURE

We share our disclosure with the authors of DNABERT-2, NT, HyenaDNA, and GenomeOcean to inform them of our findings and benchmark. Also, we highlight the potential impact on their models in our disclosure. Several of these teams have acknowledged our disclosure and confirmed awareness of the identified vulnerabilities in their models.

## K EXPERIMENT SETTING

### K.1 COMPUTATIONAL RESOURCE

We perform all experiments using 4 NVIDIA H100 GPUs with 80GB of memory and a 24-core Intel(R) Xeon(R) Gold 6338 CPU operating at 2.00 GHz.

### K.2 METRICS OF EXPERIMENTS

In our experiments, we use two core metrics to evaluate the effectiveness of adversarial attacks and the robustness of defense strategies: **Attack Success Rate (ASR)** and **Defense Success Rate (DSR)**.

**Attack Success Rate (ASR)** is defined as the relative drop in accuracy caused by the adversarial attack. Formally, let $A_{\text{clean}}$ be the model accuracy on clean inputs and $A_{\text{adv}}$ be the accuracy on adversarial inputs, then:

$$\text{ASR} = \frac{A_{\text{clean}} - A_{\text{adv}}}{A_{\text{clean}}} \times 100\%. \tag{1}$$

**Defense Success Rate (DSR)** measures the robustness achieved by applying a defense mechanism. Let $A_{\text{def}}$ be the accuracy of the defended model on adversarial inputs, then:

$$\text{DSR} = (1 - \frac{A_{\text{clean}} - A_{\text{def}}}{A_{\text{clean}}}) \times 100\%. \tag{2}$$

Without any defense mechanism,, $A_{\text{def}}$ equals the original adversarial accuracy $A_{\text{adv}}$, and thus DSR reduces to $1 - \text{ASR}$.

These metrics allow us to quantitatively assess both the impact of adversarial attacks and the degree to which defenses can mitigate that impact.

### K.3 IMPLEMENTATION

For DNABERT-2, we use the 117-million-parameter version of the model[3]. For NT, we use the 2.5-billion-parameter version of the model[4]. For NT2, we use the 100-million-parameter version of the model[5]. For HyenaDNA, we use the 4.07-million-parameter version of the model[6]. All four models represent state-of-the-art approaches for genome sequence classification tasks, consistently achieving high performance across various datasets. GenomeOcean (Zhou et al., 2025b), on the other hand, is a transformer-based model designed explicitly for genome sequence generation tasks, demonstrating superior performance compared to existing models, such as Evo (Nguyen et al., 2024a). We use the 100-million-parameter version of the model[7]. For our experiments, we fine-tuned all of these models using their official checkpoints on the datasets employed in this study.

---

[3]zhihan1996/DNABERT-2-117M
[4]InstaDeepAI/nucleotide-transformer-2.5b-multi-species
[5]InstaDeepAI/nucleotide-transformer-v2-100m-multi-species
[6]LongSafari/hyenadna-small-32k-seqlen-hf
[7]pGenomeOcean/GenomeOcean-100M

### K.4 DOWNSTREAM TASKS ACROSS DIFFERENT MODELS

We examine the downstream tasks of several genomic foundation models (GFMs), including DNABERT-2 (Zhou et al., 2024), HyenaDNA (Nguyen et al., 2024b), GenomeOcean (Zhou et al., 2025b), and Nucleotide Transformer (Dalla-Torre et al., 2024). As summarized in Table 6, these models primarily focus on classification tasks. In contrast, our analysis of the GenBench datasets (Liu et al., 2025) reveals the inclusion of regression tasks, offering a more comprehensive evaluation framework.

Table 6: **Comparison of Models (Benchmarks) and Their Tasks.**

| Model | Tasks | Classification-Only |
|---|---|---|
| DNABERT-2 | GUE (28 Classification tasks) | Yes |
| Nucleotide Transformer | Nucleotide Transformer Benchmark (18 Classification tasks) | Yes |
| HyenaDNA | GenBench (Classification-Only) + Nucleotide Transformer Benchmark | Yes |
| GenomeOcean | Classification + Generation (5 GUE Classification tasks) | No |
| GenBench | Classification + Regression (e.g., Drosophila Enhancer Activity Prediction) | No |

Table 7: **Additional Adversarial Attack Performance of the Evaluated Method.** We conduct additional experiments to assess the effectiveness of the Auto Attack and Universal Attack against target models. The table presents a comparison of target model performance before and after applying the evaluated attack. We report Attack Success Rate (ASR) as the primary evaluation metric, with variance omitted as they are all $\leq 2\%$.

| | Transformer-based | | | | Hyena-based |
|---|---|---|---|---|---|
| Attack | DNABERT-2 | NT | NT2 | GenomeOcean | HyenaDNA |
| Universal | 53.19 | 53.19 | 53.19 | 53.19 | 53.19 |
| Auto | 50.27 | 50.27 | 50.27 | 50.27 | 50.27 |

Table 8: **Additional Defense Performance Under Adversarial Attacks.** We conducted additional experiments to evaluate the performance of a defense method against adversarial attacks. The table compares the performance of target models, both with and without the evaluated defense, under Auto and Universal Attack. The Defense Success Rate (DSR) is used as the primary evaluation metric, with variance omitted as they are all $\leq 2\%$. In the table, **AT** denotes traditional adversarial training.

| | | Transformer-based | | | | Hyena-based |
|---|---|---|---|---|---|---|
| Attack Method | Defense | DNABERT-2 | NT | NT2 | GenomeOcean | HyenaDNA |
| Universal | N/A | 53.19 | 53.68 | 55.43 | 54.53 | 67.83 |
| | AT | 54.33 | 53.53 | 55.73 | 55.55 | 67.62 |
| | FreeLB | 56.56 | 55.44 | 56.67 | 55.96 | 68.07 |
| | ADFAR | 55.77 | 54.58 | 56.21 | 54.41 | 68.17 |
| Auto | N/A | 50.27 | 51.26 | 54.09 | 47.83 | 41.58 |
| | AT | 52.13 | 51.88 | 53.27 | 49.64 | 42.28 |
| | FreeLB | 51.97 | 52.21 | 54.16 | 48.02 | 42.72 |
| | ADFAR | 52.20 | 52.39 | 54.26 | 47.88 | 42.51 |

## L ADDITIONAL NUMERICAL EXPERIMENTS

### L.1 ADDITIONAL EXPERIMENTS ON UNIVERSAL AND AUTOATTACK

In this section, we present additional experimental results to further evaluate the robustness of the target models and defense strategies under adversarial attacks. Specifically, we introduce two representative attack method AutoAttack and Universal Attack, to comprehensively assess model performance. The experiments include attack success rates (ASR), defense success rates (DSR) under various defense techniques, and evaluation of model augmentation using the GenoAdv dataset.

These results provide deeper insights into the effectiveness of different strategies against adversarial perturbations and demonstrate the impact of data augmentation on adversarial robustness.

Table 9: **Additional Defense Performance Augmented with the GenoAdv Dataset.** We conduct additional experiments to evaluate the performance of a model augmented with the GenoAdv dataset against adversarial attacks. The table compares the performance of the target models, both with and without the GenoAdv dataset augmentation, under several attack methods and defense strategies. We report accuracy as the primary evaluation metric. The best results are highlighted in bold. In the table, **AT** denotes traditional adversarial training.

| Attack Method | Defense | Transformer-based | | | | Hyena-based |
|---|---|---|---|---|---|---|
| | | DNABERT-2 | NT | NT2 | GenomeOcean | HyenaDNA |
| Universal | N/A | 53.19 | 53.68 | 55.43 | 54.53 | 67.83 |
| | AT | 54.33 | 53.53 | 55.73 | 55.55 | 67.62 |
| | GenoAdv | **57.23** | **55.11** | **56.36** | **55.54** | **70.06** |
| Auto | N/A | 50.27 | 51.26 | 54.09 | 47.83 | 41.58 |
| | AT | 52.13 | 51.88 | 53.27 | 49.64 | 42.28 |
| | GenoAdv | **54.44** | **53.35** | **54.79** | **52.41** | **48.23** |

## L.2 ADVERSARIAL ATTACK VISUALIZATION ON REVERSED SAMPLES

The genomic promoter dataset contains clustered samples where the most significant promoter regions are mixed. To examine whether the salient genomic segments exhibit higher activation frequencies (as shown in Figure 4), we repeat the same experimental setup described in section 4.3 but use reversed genomic sequences. The visualization results are presented in Figure 6. We observe that for both *prom_300_tata* and *prom_300_notata*, the frequency of attack positions is reversed, indicating that regions with high attack frequency correspond to biologically significant genomic segments. The only exception is *prom_300_all*, where the frequency pattern remains unchanged despite sequence reversal. A possible explanation is that *prom_300_all* contains a much larger and more diverse dataset, with significant regions distributed across different parts of the sequence—beginning, middle, and end. When the model performs encoding, the positional attention becomes diffused, and certain heads exhibit boundary-biased "attention sink" behavior Luo et al. (2025); Hu et al. (2024); Xiao et al. (2023), over-focusing on end tokens such as EOS or padding positions. This causes adversarial perturbations near the sequence end to have disproportionately higher influence, resulting in the observed high-frequency attack positions at the tail.

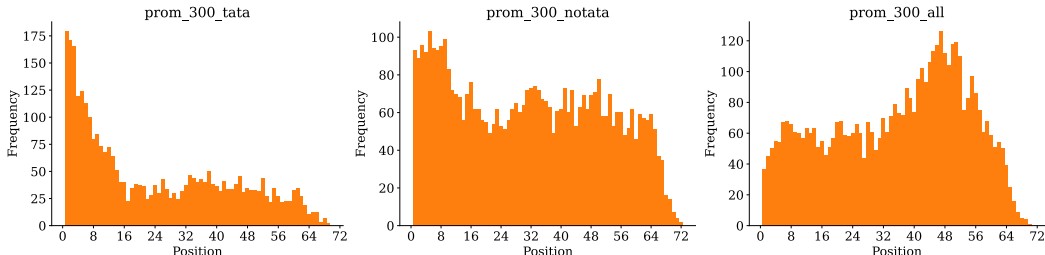

Figure 6: **Visualization examples of GFMs under adversarial attacks on reversed samples.**

## L.3 ALL RESULTS IN UNCERTAINTY ANALYSIS UNDER ADVERSARIAL ATTACKS

Our framework incorporates uncertainty quantification via Monte Carlo dropout (MC-Dropout) as a Bayesian approximation to estimate predictive posteriors. This allows us to measure model confidence, predictive entropy, as well as both aleatoric and epistemic uncertainty. We evaluate these metrics on both the original samples and adversarial samples generated by TextFooler across all GUE tasks.

Table 10: **Confidence and Entropy Comparison Under Original and Adversarial Conditions.** This table reports the original and adversarial confidence and entropy for various tasks/groups. Values are rounded to two decimal places.

| Task | Orig Conf | Adv Conf | Δ Conf | Orig Entropy | Adv Entropy | Δ Entropy |
|---|---|---|---|---|---|---|
| 0 | 0.64 | 0.75 | +0.11 | 0.65 | 0.56 | -0.09 |
| 1 | 0.88 | 0.67 | -0.21 | 0.38 | 0.64 | +0.26 |
| 2 | 0.57 | 0.57 | +0.00 | 0.68 | 0.68 | -0.00 |
| 3 | 0.68 | 0.51 | -0.17 | 0.62 | 0.69 | +0.07 |
| 4 | 0.60 | 0.51 | -0.08 | 0.68 | 0.69 | +0.02 |
| H3 | 0.76 | 0.61 | -0.15 | 0.56 | 0.67 | +0.11 |
| H3K14ac | 0.54 | 0.66 | +0.12 | 0.69 | 0.64 | -0.05 |
| H3K36me3 | 0.78 | 0.69 | -0.08 | 0.53 | 0.62 | +0.08 |
| H3K4me1 | 0.71 | 0.76 | +0.06 | 0.61 | 0.55 | -0.06 |
| H3K4me2 | 0.57 | 0.53 | -0.05 | 0.68 | 0.69 | +0.01 |
| H3K4me3 | 0.68 | 0.57 | -0.11 | 0.63 | 0.68 | +0.06 |
| H3K79me3 | 0.53 | 0.52 | -0.01 | 0.69 | 0.69 | +0.00 |
| H3K9ac | 0.64 | 0.64 | -0.00 | 0.65 | 0.66 | +0.00 |
| H4 | 0.52 | 0.78 | +0.26 | 0.69 | 0.53 | -0.16 |
| H4ac | 0.54 | 0.69 | +0.15 | 0.69 | 0.62 | -0.07 |
| prom_300_all | 0.80 | 0.54 | -0.27 | 0.50 | 0.69 | +0.19 |
| prom_300_notata | 0.78 | 0.82 | +0.05 | 0.53 | 0.47 | -0.07 |
| prom_300_tata | 0.70 | 0.56 | -0.14 | 0.61 | 0.69 | +0.08 |
| prom_core_all | 0.52 | 0.52 | +0.01 | 0.69 | 0.69 | -0.00 |
| prom_core_notata | 0.66 | 0.57 | -0.09 | 0.64 | 0.68 | +0.04 |
| prom_core_tata | 0.69 | 0.55 | -0.14 | 0.62 | 0.69 | +0.07 |
| tf0 | 0.59 | 0.73 | +0.13 | 0.68 | 0.59 | -0.09 |
| tf1 | 0.72 | 0.52 | -0.19 | 0.60 | 0.69 | +0.10 |
| tf2 | 0.71 | 0.63 | -0.08 | 0.60 | 0.66 | +0.06 |
| tf3 | 0.56 | 0.71 | +0.15 | 0.69 | 0.60 | -0.08 |
| tf4 | 0.79 | 0.76 | -0.04 | 0.51 | 0.56 | +0.04 |

As reported in Table 10, adversarial attacks induce two distinct failure modes. The first, which we term the *confused failure mode*, is characterized by a sharp drop in confidence and a corresponding increase in entropy (e.g., Task 1 and `prom_300_all`), suggesting that the model recognizes distributional anomalies and becomes uncertain. The second, more concerning, is the *overconfident failure mode*, in which the model becomes even more confident in incorrect predictions while entropy decreases (e.g., Task 0, `H4`, `H4ac`, `tf0`, and `tf3`). This behavior highlights a deeper vulnerability: under adversarial shifts, models can be confidently wrong. Our Bayesian evaluation thus reveals these internal dynamics and underscores the need for more robust defense strategies.

### L.4 ROBUSTNESS ANALYSIS UNDER CROSS-SPECIES DISTRIBUTION SHIFT

To evaluate the model's robustness against natural distribution shifts, we conducted cross-species transferability experiments using DNABERT-2. Specifically, we fine-tuned models on human transcription factor (TF) datasets and evaluated them on mouse TF datasets, and vice versa. As shown in Table 11, natural domain shifts significantly impact performance, with clean accuracy dropping to an average of approximately 48% compared to in-domain performance.

Despite this degradation due to natural shifts, the model's vulnerability to adversarial perturbations persists. When subjected to BERT-Attack, the accuracy on cross-species tasks collapses to near 0% across all tested scenarios. These results highlight a critical insight: while genomic foundation models possess limited generalization capabilities across species (natural shift), they remain catastrophically brittle to adversarial attacks (adversarial shift), regardless of the baseline performance.

Table 11: **Cross-Species Transferability and Adversarial Robustness.** This table reports the performance of models trained on one species and evaluated on another (Human ↔ Mouse) under both clean and adversarial (BERT-Attack) conditions. The *Origin Acc* column reflects robustness to natural distribution shift, while *After Attack* reflects robustness to adversarial shift.

| Direction | Task ID | Distribution Shift Setup | | Accuracy | |
|---|---|---|---|---|---|
| | | Train Domain | Test Domain | Origin Acc | After Attack |
| Train: Mouse Test: Human | TF0 | Mouse | Human | 51.00 | 0.00 |
| | TF1 | Mouse | Human | 29.00 | 0.00 |
| | TF2 | Mouse | Human | 44.00 | 0.00 |
| | TF3 | Mouse | Human | 44.00 | 1.00 |
| | TF4 | Mouse | Human | 48.00 | 0.00 |
| **Avg.** | | | | **43.20** | **0.20** |
| Train: Human Test: Mouse | TF0 | Human | Mouse | 46.00 | 1.00 |
| | TF1 | Human | Mouse | 58.00 | 1.00 |
| | TF2 | Human | Mouse | 52.00 | 0.00 |
| | TF3 | Human | Mouse | 56.00 | 0.00 |
| | TF4 | Human | Mouse | 52.00 | 0.00 |
| **Avg.** | | | | **52.80** | **0.40** |

### L.5 ABLATION STUDY: DISENTANGLING DATA AUGMENTATION FROM ADVERSARIAL DEFENSE MECHANISMS

To determine whether the effectiveness of adversarial defense strategies like FreeLB and ADFAR arises from their algorithmic principles or merely from introducing additional variations in the training data known as data augmentation, we conducted an ablation study using a *Random Mutation* baseline.

In this baseline, random perturbations were injected into 10% of the training sequences without gradient guidance or frequency-based weighting. As shown in Table 12, the Random Mutation strategy achieved a Defense Success Rate (DSR) of only 3.96%, offering minimal improvement over the undefended model. FreeLB provided a slight gain (4.34% DSR) through embedding-level gradient smoothing.

In contrast, ADFAR, which incorporates frequency-aware token randomization, achieved a substantially higher DSR of 21.84%. This difference indicates that simple stochastic augmentation is insufficient for robustifying genomic models. The enhanced performance of ADFAR is attributed to its mechanism that reduces over-reliance on specific high-frequency subsequences, effectively flattening the local loss landscape in motif-dense regions, thereby countering substitution-style attacks that exploit these biological signals.

Table 12: **Ablation Study on Defense Mechanisms.** Comparison of Defense Success Rate (DSR) between simple data augmentation and algorithmic adversarial training methods. *Random Mutation* applies 10% random noise to training data.

| Method | Mechanism | Defense Success Rate (DSR) |
|---|---|---|
| Random Mutation | Stochastic Augmentation (10% noise) | 3.96% |
| FreeLB | Embedding-level Gradient Smoothing | 4.34% |
| ADFAR | Frequency-aware Token Randomization | 21.84% |

### L.6 ALL RESULTS IN ADVERSARIAL ATTACK

This section provides a comprehensive evaluation of multiple adversarial attacks across different GFM models. We compare BertAttack, TextFooler, FIMBA, and PGD on a range of biological prediction tasks, including epigenetic marks prediction, promoter detection, and transcription factor prediction in both human and mouse datasets. The evaluated GFM models include DNABERT-2, NT, NT2, HyenaDNA, and GenomeOcean.

Table 13: **Performance Comparison of Adversarial Attacks on DNABERT-2.** This table shows the performance of all adversarial attacks on the DNABNERT-2 model. All results are evaluated using the Attack Success Rate (ASR) metric. The best result is highlighted in bold, while the second-best result is underlined.

| | Epigenetic Marks Prediction | | | | | |
| Attack | H3 | H3K14ac | H3K36me3 | H3K4me1 | H3K4me2 | H3K4me3 |
|---|---|---|---|---|---|---|
| BertAttack | **91.20** | 99.70 | 99.80 | **95.10** | **99.20** | 99.30 |
| TextFooler | 90.40 | **99.90** | **99.90** | 86.50 | **99.20** | **100.00** |
| FIMBA | 43.70 | 51.90 | 24.00 | 41.30 | 26.90 | 41.70 |
| PGD | 41.30 | 33.30 | 35.50 | 35.90 | 38.40 | 31.80 |

| | Epigenetic Marks Prediction | | | | Promoter Detection (300bp) | | |
| Attack | H3K79me3 | H3K9ac | H4 | H4ac | all | notata | tata |
|---|---|---|---|---|---|---|---|
| BertAttack | 97.50 | **98.00** | **96.60** | **100.00** | **83.70** | **92.70** | 96.50 |
| TextFooler | **99.40** | 96.20 | 96.00 | 94.20 | 71.80 | 28.30 | **97.00** |
| FIMBA | 24.40 | 43.80 | 36.60 | 50.60 | 58.30 | 14.90 | 87.10 |
| PGD | 41.40 | 39.30 | 36.20 | 46.10 | 45.60 | 43.50 | 42.90 |

| | Transcription Factor Prediction (Human) | | | | | Core Promoter Detection | | |
| Attack | tf0 | tf1 | tf2 | tf3 | tf4 | all | notata | tata |
|---|---|---|---|---|---|---|---|---|
| BertAttack | 96.80 | 97.60 | 99.80 | 90.20 | 97.40 | **99.20** | **99.30** | **98.90** |
| TextFooler | 96.40 | **98.00** | **99.40** | **91.30** | **98.80** | 97.40 | 97.10 | 92.00 |
| FIMBA | 50.00 | 34.10 | 55.60 | 25.40 | 45.30 | 44.00 | 32.10 | 28.20 |
| PGD | 36.60 | 32.30 | 35.60 | 34.80 | 41.00 | 35.10 | 34.10 | 35.80 |

| | Transcription Factor Prediction (Mouse) | | | | |
| Attack | 0 | 1 | 2 | 3 | 4 |
|---|---|---|---|---|---|
| BertAttack | 93.40 | **96.40** | 96.20 | 90.90 | **96.90** |
| TextFooler | **94.20** | 94.50 | **97.20** | **92.40** | 94.20 |
| FIMBA | 46.40 | 3.10 | 43.30 | 46.40 | 39.50 |
| PGD | 43.50 | 38.80 | 35.10 | 45.40 | 36.00 |

Table 14: **Performance Comparison of Adversarial Attacks on HyenaDNA.** This table shows the performance of all adversarial attacks on the HyenaDNA model. All results are evaluated using the Attack Success Rate (ASR) metric. The best result is highlighted in bold, while the second-best result is underlined.

| | Epigenetic Marks Prediction | | | | | |
| Attack | H3 | H3K14ac | H3K36me3 | H3K4me1 | H3K4me2 | H3K4me3 |
|---|---|---|---|---|---|---|
| BertAttack | **100.00** | **100.00** | **100.00** | **99.06** | **100.00** | **100.00** |
| TextFooler | **100.00** | **100.00** | **100.00** | 92.70 | **100.00** | 91.14 |
| FIMBA | 46.27 | 3.17 | 3.51 | 16.13 | 14.81 | 8.20 |
| PGD | 10.70 | 6.70 | 91.14 | 5.11 | 90.68 | 4.45 |

| | Epigenetic Marks Prediction | | | | Promoter Detection (300bp) | | |
| Attack | H3K79me3 | H3K9ac | H4 | H4ac | all | notata | tata |
|---|---|---|---|---|---|---|---|
| BertAttack | **100.00** | **100.00** | **100.00** | **100.00** | **100.00** | 97.06 | **100.00** |
| TextFooler | 35.79 | 41.68 | **100.00** | 99.19 | 46.49 | **99.19** | 92.85 |
| FIMBA | 25.86 | 38.10 | 18.18 | 35.48 | 48.68 | 31.17 | 41.67 |
| PGD | 7.04 | 12.23 | 22.12 | 2.58 | 25.13 | 92.41 | 93.72 |

| | Transcription Factor Prediction (Human) | | | | | Core Promoter Detection | | |
| Attack | tf0 | tf1 | tf2 | tf3 | tf4 | all | notata | tata |
|---|---|---|---|---|---|---|---|---|
| BertAttack | **100.00** | 99.88 | **100.00** | **100.00** | 98.81 | **100.00** | **100.00** | **100.00** |
| TextFooler | **100.00** | **100.00** | **100.00** | **100.00** | **100.00** | **100.00** | **100.00** | **100.00** |
| FIMBA | 38.16 | 35.71 | 31.94 | 26.39 | 48.86 | 34.15 | 32.14 | 33.33 |
| PGD | 90.42 | 92.86 | 93.24 | 90.70 | 96.65 | 24.47 | 12.25 | 93.59 |

| | Transcription Factor Prediction (Mouse) | | | | |
| Attack | 0 | 1 | 2 | 3 | 4 |
|---|---|---|---|---|---|
| BertAttack | **100.00** | 99.97 | **100.00** | **100.00** | 98.79 |
| TextFooler | 0.74 | **100.00** | **100.00** | **100.00** | **100.00** |
| FIMBA | 40.79 | 40.22 | 36.59 | 32.84 | 26.67 |
| PGD | 0.00 | 4.35 | 2.65 | 90.99 | 90.18 |

Table 15: **Performance Comparison of Adversarial Attacks on NT.** This table shows the performance of all adversarial attacks on the Nucleotide Transformer (NT) model. All results are evaluated using the Attack Success Rate (ASR) metric. The best result is highlighted in bold, while the second-best result is underlined.

| | Epigenetic Marks Prediction | | | | | |
|---|---|---|---|---|---|---|
| Attack | H3 | H3K14ac | H3K36me3 | H3K4me1 | H3K4me2 | H3K4me3 |
| BertAttack | **99.92** | **100.00** | **100.00** | **100.00** | **100.00** | **100.00** |
| TextFooler | 66.23 | **100.00** | 92.29 | 97.32 | **100.00** | **100.00** |
| FIMBA | 55.13 | 42.65 | 25.00 | 22.06 | 39.06 | 31.67 |
| PGD | 38.53 | 38.45 | 39.11 | 36.16 | 36.93 | 25.25 |

| | Epigenetic Marks Prediction | | | | Promoter Detection (300bp) | | |
|---|---|---|---|---|---|---|---|
| Attack | H3K79me3 | H3K9ac | H4 | H4ac | all | notata | tata |
| BertAttack | **100.00** | **100.00** | **99.24** | **100.00** | **100.00** | **100.00** | **100.00** |
| TextFooler | **100.00** | **100.00** | 90.70 | 89.24 | 99.19 | **100.00** | 91.20 |
| FIMBA | 30.77 | 36.36 | 58.89 | 32.20 | 57.45 | 44.90 | 46.51 |
| PGD | 40.91 | 20.45 | 38.24 | 39.11 | 36.14 | 35.47 | 36.70 |

| | Transcription Factor Prediction (Human) | | | | | Core Promoter Detection | | |
|---|---|---|---|---|---|---|---|---|
| Attack | tf0 | tf1 | tf2 | tf3 | tf4 | all | notata | tata |
| BertAttack | **100.00** | **100.00** | 99.72 | **100.00** | **100.00** | 99.76 | 99.55 | 99.27 |
| TextFooler | **100.00** | **100.00** | **100.00** | **100.00** | 95.39 | **100.00** | **100.00** | **100.00** |
| FIMBA | 37.33 | 41.98 | 30.99 | 20.90 | 43.04 | 33.80 | 35.23 | 42.86 |
| PGD | 46.85 | 48.61 | 34.57 | 39.56 | 53.13 | 38.24 | 39.04 | 57.08 |

| | Transcription Factor Prediction (Mouse) | | | | |
|---|---|---|---|---|---|
| Attack | 0 | 1 | 2 | 3 | 4 |
| BertAttack | **100.00** | **99.66** | 99.46 | **100.00** | **100.00** |
| TextFooler | **100.00** | 92.47 | **100.00** | **100.00** | **100.00** |
| FIMBA | 35.71 | 51.06 | 39.02 | 16.36 | 28.13 |
| PGD | 26.10 | 41.97 | 37.61 | 45.96 | 23.91 |

Table 16: **Performance Comparison of Adversarial Attacks on NT2.** This table shows the performance of all adversarial attacks on the Nucleotide Transformer 2 (NT2) model. All results are evaluated using the Attack Success Rate (ASR) metric. The best result is highlighted in bold, while the second-best result is underlined.

| | Epigenetic Marks Prediction | | | | | |
|---|---|---|---|---|---|---|
| Attack | H3 | H3K14ac | H3K36me3 | H3K4me1 | H3K4me2 | H3K4me3 |
| BertAttack | 98.42 | 99.62 | 99.91 | 99.66 | **100.00** | **100.00** |
| TextFooler | **100.00** | **100.00** | **100.00** | **100.00** | **100.00** | **100.00** |
| FIMBA | 27.38 | 22.08 | 34.48 | 30.26 | 23.53 | 39.71 |
| PGD | 43.55 | 35.86 | 16.13 | 11.19 | 38.99 | 11.95 |

| | Epigenetic Marks Prediction | | | | Promoter Detection (300bp) | | |
|---|---|---|---|---|---|---|---|
| Attack | H3K79me3 | H3K9ac | H4 | H4ac | all | notata | tata |
| BertAttack | **100.00** | 99.53 | 99.45 | **100.00** | 99.70 | **95.35** | 99.47 |
| TextFooler | **100.00** | **100.00** | **100.00** | **100.00** | **100.00** | 88.59 | **100.00** |
| FIMBA | 6.02 | 62.03 | 23.08 | 25.61 | 59.60 | 9.09 | 51.58 |
| PGD | 34.78 | 38.82 | 32.60 | 38.35 | 35.34 | 32.95 | 18.03 |

| | Transcription Factor Prediction (Human) | | | | | Core Promoter Detection | | |
|---|---|---|---|---|---|---|---|---|
| Attack | tf0 | tf1 | tf2 | tf3 | tf4 | all | notata | tata |
| BertAttack | **100.00** | **100.00** | **100.00** | **99.83** | **100.00** | 99.63 | 99.31 | **99.64** |
| TextFooler | **100.00** | **100.00** | 88.84 | 99.80 | **100.00** | **99.81** | **100.00** | 40.23 |
| FIMBA | 44.71 | 28.95 | 37.18 | 33.75 | 50.55 | 45.35 | 34.48 | 44.79 |
| PGD | 50.82 | 65.69 | 45.11 | 36.52 | 63.40 | 11.81 | 37.73 | 37.70 |

| | Transcription Factor Prediction (Mouse) | | | | |
|---|---|---|---|---|---|
| Attack | 0 | 1 | 2 | 3 | 4 |
| BertAttack | **100.00** | 99.59 | **99.49** | **100.00** | **100.00** |
| TextFooler | 99.78 | **99.82** | 95.74 | **100.00** | 97.84 |
| FIMBA | 50.00 | 42.71 | 40.70 | 38.89 | 42.50 |
| PGD | 38.69 | 40.22 | 15.00 | 41.88 | 21.56 |

Table 17: **Performance Comparison of Adversarial Attacks on GenomeOcean.** This table shows the performance of all adversarial attacks on the GenomeOcean model. All results are evaluated using the Attack Success Rate (ASR) metric. The best result is highlighted in bold, while the second-best result is underlined.

| | Epigenetic Marks Prediction | | | | | |
|---|---|---|---|---|---|---|
| Attack | H3 | H3K14ac | H3K36me3 | H3K4me1 | H3K4me2 | H3K4me3 |
| BertAttack | **100.00** | 99.60 | 99.97 | **100.00** | 99.95 | 99.97 |
| TextFooler | 99.78 | **100.00** | **100.00** | **100.00** | **100.00** | **100.00** |
| FIMBA | 45.88 | 36.14 | 24.10 | 49.35 | 53.73 | 51.95 |
| PGD | 47.74 | 42.41 | 41.11 | 48.82 | 38.28 | 45.57 |

| | Epigenetic Marks Prediction | | | | Promoter Detection (300bp) | | |
|---|---|---|---|---|---|---|---|
| Attack | H3K79me3 | H3K9ac | H4 | H4ac | all | notata | tata |
| BertAttack | 98.75 | **100.00** | **98.18** | 98.51 | 99.65 | **100.00** | 97.71 |
| TextFooler | **100.00** | **100.00** | 88.89 | **100.00** | **99.87** | **100.00** | **100.00** |
| FIMBA | 43.37 | 21.52 | 35.16 | 68.67 | 59.78 | 36.36 | 28.57 |
| PGD | 44.12 | 48.49 | 43.45 | 18.72 | 53.34 | 41.15 | 35.22 |

| | Transcription Factor Prediction (Human) | | | | | Core Promoter Detection | | |
|---|---|---|---|---|---|---|---|---|
| Attack | tf0 | tf1 | tf2 | tf3 | tf4 | all | notata | tata |
| BertAttack | **100.00** | **100.00** | **99.89** | 99.60 | 99.94 | 99.83 | 99.91 | 99.81 |
| TextFooler | **100.00** | **100.00** | 99.88 | **99.85** | **100.00** | **100.00** | **100.00** | **100.00** |
| FIMBA | 46.91 | 31.65 | 49.37 | 39.39 | 45.88 | 42.68 | 31.33 | 38.96 |
| PGD | 22.98 | 22.98 | 23.95 | 33.33 | 22.06 | 41.39 | 32.15 | 39.66 |

| | Transcription Factor Prediction (Mouse) | | | | |
|---|---|---|---|---|---|
| Attack | 0 | 1 | 2 | 3 | 4 |
| BertAttack | **100.00** | 99.83 | 98.95 | 98.83 | **100.00** |
| TextFooler | **100.00** | **99.89** | **100.00** | **99.90** | **100.00** |
| FIMBA | 1.16 | 53.68 | 34.83 | 57.65 | 39.47 |
| PGD | 43.36 | 23.68 | 24.94 | 32.90 | 38.91 |

Table 18: **Performance Comparison of Adversarial Defense on DNABERT-2.** This table shows the performance of all adversarial defense on the DNABERT-2 model. All results are evaluated using the Defense Success Rate (DSR) metric. The best result is highlighted in bold, while the second-best result is underlined.

| Attack | Defense | Epigenetic Marks Prediction | | | | | |
| | | H3 | H3K14ac | H3K36me3 | H3K4me1 | H3K4me2 | H3K4me3 |
|---|---|---|---|---|---|---|---|
| PGD | FreeLB | 56.17 | 65.68 | 66.22 | 63.10 | 72.38 | 63.92 |
| | ADFAR | **64.32** | 63.55 | 65.51 | 62.01 | 74.57 | **64.58** |
| | AT | 54.87 | **77.97** | **69.08** | **72.55** | **82.38** | 61.01 |
| BertAttack | FreeLB | 5.10 | 0.00 | 1.16 | 0.00 | 1.19 | 10.00 |
| | ADFAR | **100.00** | 0.00 | **10.10** | 0.00 | 2.08 | **94.23** |
| | AT | 4.76 | 0.00 | 0.00 | 0.00 | **2.86** | 0.00 |
| TextFooler | FreeLB | 33.88 | 0.11 | 0.00 | 0.00 | 0.00 | 0.00 |
| | ADFAR | **42.28** | 0.00 | 0.00 | 0.00 | 0.00 | **0.22** |
| | AT | 41.25 | **0.12** | **0.12** | 0.00 | **1.88** | 0.00 |

| Attack | Defense | Epigenetic Marks Prediction | | | | Promoter Detection (300bp) | | |
| | | H3K79me3 | H3K9ac | H4 | H4ac | all | notata | tata |
|---|---|---|---|---|---|---|---|---|
| PGD | FreeLB | 61.47 | **63.44** | 60.84 | **67.58** | 55.93 | 56.01 | 58.74 |
| | ADFAR | 62.08 | 55.82 | 65.56 | 62.38 | **70.01** | **65.59** | **64.26** |
| | AT | **62.91** | 60.92 | **73.12** | 59.48 | 63.67 | 51.98 | 49.74 |
| BertAttack | FreeLB | 0.00 | 1.08 | 6.19 | 0.00 | 0.00 | 1.00 | **9.28** |
| | ADFAR | 0.00 | **8.42** | 0.00 | **25.00** | **4.08** | **100.00** | 7.69 |
| | AT | **4.55** | 4.29 | **15.62** | 0.00 | 2.04 | 19.59 | 8.75 |
| TextFooler | FreeLB | 0.00 | 0.00 | 34.68 | 0.00 | 0.00 | 3.04 | 73.16 |
| | ADFAR | 0.00 | 0.00 | **76.39** | **4.74** | **8.42** | **100.00** | **88.83** |
| | AT | **1.28** | **5.57** | 38.16 | 0.00 | 0.00 | 28.97 | 75.63 |

| Attack | Defense | Transcription Factor Prediction (Human) | | | | | Core Promoter Detection | | |
| | | tf0 | tf1 | tf2 | tf3 | tf4 | all | notata | tata |
|---|---|---|---|---|---|---|---|---|---|
| PGD | FreeLB | **66.17** | **72.23** | 73.21 | **66.79** | **65.54** | 73.30 | 69.31 | **64.18** |
| | ADFAR | 64.78 | 64.38 | 56.85 | 56.18 | 61.97 | 60.61 | 67.32 | 59.56 |
| | AT | 64.44 | 64.76 | **77.58** | 59.93 | 57.08 | **74.31** | **76.35** | 62.18 |
| BertAttack | FreeLB | **10.20** | 0.00 | 10.00 | **2.15** | 2.27 | 0.00 | 0.00 | 0.00 |
| | ADFAR | 0.00 | 0.00 | 0.00 | 0.00 | **100.00** | **27.08** | **7.07** | 0.00 |
| | AT | 0.00 | 0.00 | **10.34** | 0.00 | 0.00 | 1.20 | 1.14 | **1.10** |
| TextFooler | FreeLB | 0.22 | 0.00 | 0.00 | 0.34 | 0.71 | 0.00 | 1.01 | 72.85 |
| | ADFAR | 0.00 | 0.00 | **6.29** | **100.00** | 2.41 | **26.29** | **1.61** | **97.97** |
| | AT | **0.98** | 0.00 | 0.24 | 0.13 | **3.17** | 0.00 | 0.66 | 75.18 |

| Attack | Defense | Transcription Factor Prediction (Mouse) | | | | |
| | | 0 | 1 | 2 | 3 | 4 |
|---|---|---|---|---|---|---|
| PGD | FreeLB | 57.93 | 70.40 | 56.17 | 57.29 | **61.82** |
| | ADFAR | **69.44** | 64.73 | **60.40** | **62.41** | 61.54 |
| | AT | 55.61 | **73.15** | **73.22** | 53.08 | 56.45 |
| BertAttack | FreeLB | 20.62 | 4.12 | **9.00** | **17.35** | 2.20 |
| | ADFAR | **44.44** | **27.27** | 0.00 | 10.42 | **100.00** |
| | AT | 5.49 | 10.20 | 6.82 | 5.71 | 1.10 |
| TextFooler | FreeLB | 65.90 | 0.00 | 85.89 | 89.98 | 16.28 |
| | ADFAR | 67.49 | **17.54** | **91.92** | **96.23** | **26.15** |
| | AT | **68.2** | 6.18 | 87.45 | 92.45 | 17.54 |

Table 19: **Performance Comparison of Adversarial Defense on GenomeOcean.** This table shows the performance of all adversarial defense on the GenomeOcean model. All results are evaluated using the Defense Success Rate (DSR) metric. The best result is highlighted in bold, while the second-best result is underlined.

| Attack | Defense | Epigenetic Marks Prediction | | | | | |
| --- | --- | --- | --- | --- | --- | --- | --- |
| | | H3 | H3K14ac | H3K36me3 | H3K4me1 | H3K4me2 | H3K4me3 |
| PGD | FreeLB | **58.51** | 50.75 | 52.96 | 55.52 | 58.13 | **56.48** |
| | ADFAR | 54.75 | **66.59** | 49.43 | **68.20** | **69.17** | 50.24 |
| | AT | 57.40 | 55.78 | **59.35** | 49.87 | 64.69 | 52.15 |
| BertAttack | FreeLB | **2.04** | **8.60** | **3.19** | 0.00 | 0.00 | 0.00 |
| | ADFAR | 0.00 | 0.00 | 0.00 | 0.00 | 0.00 | 0.00 |
| | AT | 0.22 | 4.45 | 0.13 | **0.04** | **0.22** | 0.00 |
| TextFooler | FreeLB | 0.00 | 0.00 | 0.00 | 0.00 | 0.00 | 0.00 |
| | ADFAR | 0.00 | 0.00 | 0.00 | 0.00 | 0.00 | 0.00 |
| | AT | **33.75** | 0.00 | 0.00 | 0.00 | 0.00 | 0.00 |

| Attack | Defense | Epigenetic Marks Prediction | | | | Promoter Detection (300bp) | | |
| --- | --- | --- | --- | --- | --- | --- | --- | --- |
| | | H3K79me3 | H3K9ac | H4 | H4ac | all | notata | tata |
| PGD | FreeLB | **57.31** | 55.04 | 56.79 | **93.99** | 45.78 | 52.47 | 63.83 |
| | ADFAR | 51.14 | 46.38 | **61.72** | 86.29 | 52.74 | 51.28 | 64.24 |
| | AT | 56.04 | **55.60** | 56.85 | 92.58 | **53.46** | **65.77** | **66.48** |
| BertAttack | FreeLB | **6.12** | **22.99** | **24.24** | **1.05** | 0.00 | 0.00 | 0.00 |
| | ADFAR | 0.00 | 0.00 | 0.00 | 0.00 | 0.00 | **3.77** | 0.00 |
| | AT | 1.05 | 1.69 | 1.31 | 0.75 | 0.00 | 0.04 | 0.00 |
| TextFooler | FreeLB | 0.00 | 0.00 | 35.25 | 0.00 | 0.00 | 0.00 | 73.8 |
| | ADFAR | 0.00 | 0.00 | 0.51 | 0.00 | 0.00 | **100.00** | **100.00** |
| | AT | 0.00 | 0.00 | **37.13** | 0.00 | 0.00 | 0.00 | 74.10 |

| Attack | Defense | Transcription Factor Prediction (Human) | | | | | Core Promoter Detection | | |
| --- | --- | --- | --- | --- | --- | --- | --- | --- | --- |
| | | tf0 | tf1 | tf2 | tf3 | tf4 | all | notata | tata |
| PGD | FreeLB | **96.51** | 91.09 | 91.18 | 67.74 | 91.79 | **70.92** | 66.86 | 57.31 |
| | ADFAR | 92.09 | **97.83** | 93.73 | 67.07 | **96.94** | 57.38 | 58.62 | 55.30 |
| | AT | 91.54 | 93.95 | **94.37** | **68.14** | 96.53 | 60.82 | **69.29** | **61.32** |
| BertAttack | FreeLB | 0.00 | 0.00 | 0.00 | 0.00 | **1.00** | **2.15** | 0.00 | **1.01** |
| | ADFAR | 0.00 | 0.00 | 0.00 | 0.00 | 0.00 | 1.85 | 0.00 | 0.00 |
| | AT | 0.00 | 0.00 | 0.00 | 0.00 | 0.00 | 0.76 | **0.80** | 0.18 |
| TextFooler | FreeLB | 0.00 | 0.00 | 0.00 | 0.00 | 0.00 | 0.00 | 0.00 | 73.13 |
| | ADFAR | **100.00** | **100.00** | **100.00** | **100.00** | **100.00** | 0.00 | 0.00 | 0.10 |
| | AT | 0.00 | 0.52 | 0.00 | 0.00 | 0.42 | 0.00 | 0.00 | **74.49** |

| Attack | Defense | Transcription Factor Prediction (Mouse) | | | | |
| --- | --- | --- | --- | --- | --- | --- |
| | | 0 | 1 | 2 | 3 | 4 |
| PGD | FreeLB | 57.25 | **73.37** | 68.87 | 67.39 | 57.16 |
| | ADFAR | 55.60 | 69.74 | **69.72** | **68.53** | 57.96 |
| | AT | **58.48** | 70.22 | 48.47 | 61.68 | **58.82** |
| BertAttack | FreeLB | 0.00 | 1.05 | 2.00 | 1.00 | 0.00 |
| | ADFAR | 0.00 | **25.00** | 0.00 | 0.00 | 0.00 |
| | AT | 0.00 | 0.00 | **2.02** | **2.00** | 0.00 |
| TextFooler | FreeLB | 64.44 | 0.00 | 85.73 | 89.57 | 28.76 |
| | ADFAR | **100.00** | **100.00** | **100.00** | **100.00** | **100.00** |
| | AT | 65.98 | 1.65 | 85.47 | 90.03 | 17.63 |

Table 20: **Performance Comparison of Adversarial Defense on NT.** This table shows the performance of all adversarial defense on the Nucleotide Transformer (NT) model. All results are evaluated using the Defense Success Rate (DSR) metric. The best result is highlighted in bold, while the second-best result is underlined.

| Attack | Defense | Epigenetic Marks Prediction | | | | | |
| --- | --- | --- | --- | --- | --- | --- | --- |
| | | H3 | H3K14ac | H3K36me3 | H3K4me1 | H3K4me2 | H3K4me3 |
| PGD | FreeLB | 87.79 | 84.45 | 80.44 | 85.20 | 84.35 | 74.08 |
| | ADFAR | 54.65 | 53.07 | 50.08 | 57.23 | 54.72 | 57.95 |
| | AT | **92.35** | **86.74** | **82.02** | **86.80** | **87.54** | **75.02** |
| BertAttack | FreeLB | **7.14** | 0.00 | 0.00 | **1.18** | 0.00 | 0.00 |
| | ADFAR | 2.04 | 0.00 | 0.00 | 0.00 | **13.56** | 0.00 |
| | AT | 0.22 | 0.00 | 0.00 | 0.00 | 0.00 | 0.00 |
| TextFooler | FreeLB | 25.69 | 23.10 | 10.30 | 12.40 | 20.00 | **9.54** |
| | ADFAR | 0.00 | **100.00** | **62.70** | **12.90** | 9.35 | 7.33 |
| | AT | **47.68** | 24.97 | 12.31 | 9.39 | **47.97** | 7.99 |

| Attack | Defense | Epigenetic Marks Prediction | | | | Promoter Detection (300bp) | | |
| --- | --- | --- | --- | --- | --- | --- | --- | --- |
| | | H3K79me3 | H3K9ac | H4 | H4ac | all | notata | tata |
| PGD | FreeLB | **85.64** | **83.36** | **89.65** | **84.87** | 93.93 | 95.41 | **99.34** |
| | ADFAR | 53.70 | 61.02 | 59.09 | 59.92 | 52.09 | 51.25 | 57.63 |
| | AT | 84.74 | 81.22 | 82.81 | 81.39 | **94.26** | **96.97** | 90.45 |
| BertAttack | FreeLB | 0.00 | 0.00 | 2.06 | 0.00 | 0.00 | 0.00 | 2.02 |
| | ADFAR | **6.52** | 0.00 | **2.17** | 0.00 | 0.00 | **43.75** | **11.76** |
| | AT | 0.00 | 0.00 | 2.04 | 0.00 | 0.00 | 1.02 | 0.00 |
| TextFooler | FreeLB | **22.17** | 41.03 | 62.86 | 35.14 | 35.79 | 31.25 | 85.07 |
| | ADFAR | 2.55 | **100.00** | **72.48** | **42.77** | **49.20** | **69.14** | **91.32** |
| | AT | 13.34 | 24.61 | 53.74 | 23.82 | 35.97 | 34.56 | 82.09 |

| Attack | Defense | Transcription Factor Prediction (Human) | | | | | Core Promoter Detection | | |
| --- | --- | --- | --- | --- | --- | --- | --- | --- | --- |
| | | tf0 | tf1 | tf2 | tf3 | tf4 | all | notata | tata |
| PGD | FreeLB | 57.58 | 55.28 | 72.30 | **82.95** | 48.23 | **85.07** | 89.47 | 39.77 |
| | ADFAR | **96.49** | **97.26** | **92.94** | 59.67 | **96.92** | 54.34 | 56.28 | **95.70** |
| | AT | 84.21 | 59.95 | 66.17 | 62.06 | 64.31 | 81.73 | 91.93 | 38.15 |
| BertAttack | FreeLB | 0.00 | 0.00 | 0.00 | 0.00 | 0.00 | 1.04 | **1.06** | **1.02** |
| | ADFAR | 0.00 | 0.00 | 0.00 | **5.66** | 0.00 | 0.00 | 0.00 | 0.00 |
| | AT | 0.00 | 0.00 | 0.00 | 0.00 | 0.00 | **1.15** | 0.00 | **1.02** |
| TextFooler | FreeLB | 36.03 | 34.17 | 32.44 | 28.15 | 38.83 | 44.54 | 47.10 | 89.26 |
| | ADFAR | **100.00** | **60.02** | **75.00** | **99.58** | **89.74** | **64.10** | **100.00** | **100.00** |
| | AT | 43.85 | 41.76 | 28.65 | 44.43 | 37.16 | 34.18 | 35.66 | 86.49 |

| Attack | Defense | Transcription Factor Prediction (Mouse) | | | | |
| --- | --- | --- | --- | --- | --- | --- |
| | | 0 | 1 | 2 | 3 | 4 |
| PGD | FreeLB | 74.60 | 98.03 | 86.32 | **70.60** | 75.08 |
| | ADFAR | 56.57 | 55.57 | 53.62 | 52.30 | 59.28 |
| | AT | **76.37** | **99.44** | **99.46** | 34.72 | **75.64** |
| BertAttack | FreeLB | 0.00 | 0.00 | 0.00 | **2.02** | 0.00 |
| | ADFAR | 0.00 | **41.07** | **2.13** | 0.00 | 0.00 |
| | AT | 0.00 | 0.13 | 0.00 | 0.00 | 0.00 |
| TextFooler | FreeLB | 75.28 | 58.13 | 92.57 | 93.98 | 31.72 |
| | ADFAR | **85.24** | **83.82** | **97.60** | **100.00** | **69.05** |
| | AT | 72.34 | 56.08 | 89.80 | 94.44 | 31.63 |

Table 21: **Performance Comparison of Adversarial Defense on NT2.** This table shows the performance of all adversarial defense on the Nucleotide Transformer-2 (NT2) model. All results are evaluated using the Defense Success Rate (DSR) metric. The best result is highlighted in bold, while the second-best result is underlined.

| Attack | Defense | Epigenetic Marks Prediction | | | | | |
| | | H3 | H3K14ac | H3K36me3 | H3K4me1 | H3K4me2 | H3K4me3 |
|---|---|---|---|---|---|---|---|
| PGD | FreeLB | 89.10 | 80.99 | 79.18 | 84.75 | **76.23** | 76.21 |
| | ADFAR | 86.57 | 73.38 | 77.85 | 77.95 | 55.52 | 67.38 |
| | AT | **97.61** | **82.31** | **83.20** | **86.88** | 75.67 | **77.66** |
| BertAttack | FreeLB | **2.02** | 0.00 | 0.00 | 0.00 | 0.00 | 0.00 |
| | ADFAR | 0.00 | **5.97** | **1.67** | 0.00 | 0.00 | 0.00 |
| | AT | 0.00 | 0.00 | 0.00 | 0.00 | 0.00 | 0.00 |
| TextFooler | FreeLB | 33.23 | 0.00 | 0.00 | 0.00 | 0.00 | 0.00 |
| | ADFAR | **49.57** | 0.00 | 0.00 | 0.00 | 0.00 | 0.00 |
| | AT | 35.70 | 0.00 | 0.00 | 0.00 | 0.00 | 0.00 |

| Attack | Defense | Epigenetic Marks Prediction | | | | Promoter Detection (300bp) | | |
| | | H3K79me3 | H3K9ac | H4 | H4ac | all | notata | tata |
|---|---|---|---|---|---|---|---|---|
| PGD | FreeLB | **89.83** | 84.55 | **99.44** | 79.34 | **94.93** | **91.57** | **94.00** |
| | ADFAR | 89.28 | 73.77 | 74.60 | 73.34 | 61.27 | 57.61 | 70.18 |
| | AT | 89.43 | **86.33** | 96.76 | **83.78** | 93.74 | 90.42 | 85.16 |
| BertAttack | FreeLB | 0.00 | 0.00 | 3.12 | 0.00 | 0.00 | 0.00 | **1.00** |
| | ADFAR | **18.18** | 0.00 | **4.26** | 0.00 | 0.00 | **18.75** | 0.00 |
| | AT | 0.00 | 0.00 | 1.02 | 0.00 | 0.00 | 0.00 | 0.00 |
| TextFooler | FreeLB | 0.00 | **0.11** | 35.29 | 0.00 | **0.71** | 0.00 | 73.73 |
| | ADFAR | 0.00 | 0.00 | **72.82** | 0.00 | 0.00 | **0.71** | **76.05** |
| | AT | 0.00 | 0.00 | 35.22 | 0.00 | 0.00 | 0.00 | 74.33 |

| Attack | Defense | Transcription Factor Prediction (Human) | | | | | Core Promoter Detection | | |
| | | tf0 | tf1 | tf2 | tf3 | tf4 | all | notata | tata |
|---|---|---|---|---|---|---|---|---|---|
| PGD | FreeLB | **92.86** | **94.01** | **82.76** | 84.25 | **97.22** | **91.26** | **91.33** | **99.82** |
| | ADFAR | 73.62 | 68.87 | 73.46 | 71.17 | 75.97 | 73.68 | 78.40 | 62.35 |
| | AT | 61.98 | 68.87 | 61.98 | **87.24** | 94.12 | 88.43 | 76.66 | 55.54 |
| BertAttack | FreeLB | 0.00 | 0.00 | 0.00 | **2.22** | 0.00 | 0.00 | 0.00 | 0.00 |
| | ADFAR | **51.06** | **60.38** | 0.00 | 0.00 | **2.04** | 0.00 | 0.00 | 0.00 |
| | AT | 0.00 | 4.00 | **1.00** | 0.00 | 0.00 | 0.00 | 0.00 | **1.00** |
| TextFooler | FreeLB | 0.00 | 0.00 | 0.00 | 0.00 | 0.00 | 0.00 | 0.00 | 72.76 |
| | ADFAR | 0.00 | 0.00 | 0.00 | 0.00 | 0.00 | 0.00 | 0.00 | **85.48** |
| | AT | 0.00 | 0.00 | **0.15** | **0.12** | 0.00 | 0.00 | 0.00 | 77.81 |

| Attack | Defense | Transcription Factor Prediction (Mouse) | | | | |
| | | 0 | 1 | 2 | 3 | 4 |
|---|---|---|---|---|---|---|
| PGD | FreeLB | **88.71** | **99.19** | 97.29 | 81.65 | **81.44** |
| | ADFAR | 77.22 | 74.06 | **99.56** | 66.95 | 61.05 |
| | AT | 74.61 | 97.07 | **99.56** | **86.09** | 51.29 |
| BertAttack | FreeLB | 0.00 | **4.04** | **2.00** | **4.08** | 0.00 |
| | ADFAR | **1.92** | 0.00 | 0.00 | 0.00 | **16.67** |
| | AT | 0.00 | 4.00 | 1.00 | 0.00 | 0.00 |
| TextFooler | FreeLB | 63.98 | 0.00 | 85.96 | 89.66 | 16.67 |
| | ADFAR | **77.00** | 0.00 | 86.34 | **94.90** | **29.07** |
| | AT | 67.30 | 0.20 | **86.69** | 92.44 | 22.71 |

Table 22: **Performance Comparison of Adversarial Defense on HyenaDNA.** This table shows the performance of all adversarial defense on the HyenaDNA model. All results are evaluated using the Defense Success Rate (DSR) metric. The best result is highlighted in bold, while the second-best result is underlined.

| Attack | Defense | Epigenetic Marks Prediction | | | | | |
|--------|---------|------|--------|---------|--------|--------|--------|
| | | H3 | H3K14ac | H3K36me3 | H3K4me1 | H3K4me2 | H3K4me3 |
| PGD | FreeLB | 76.72 | 70.87 | 98.19 | 91.86 | 96.22 | 85.29 |
| | ADFAR | **88.44** | 74.31 | 85.63 | **94.41** | **98.83** | 84.20 |
| | AT | **88.44** | **84.26** | **99.36** | 86.77 | 91.96 | **87.48** |
| BertAttack | FreeLB | 0.00 | 0.00 | 0.00 | 0.00 | 0.00 | 0.00 |
| | ADFAR | 0.00 | 0.00 | 0.00 | 0.00 | 0.00 | 0.00 |
| | AT | 0.00 | 0.00 | 0.00 | 0.00 | 0.00 | 0.00 |
| TextFooler | FreeLB | **100.00** | 98.08 | 71.00 | **75.21** | 53.82 | **100.00** |
| | ADFAR | **100.00** | **99.77** | 30.70 | 50.62 | 29.01 | 97.75 |
| | AT | **100.00** | 84.18 | **95.87** | 50.68 | **64.87** | 80.81 |

| Attack | Defense | Epigenetic Marks Prediction | | | | Promoter Detection (300bp) | | |
|--------|---------|----------|--------|--------|--------|------|--------|--------|
| | | H3K79me3 | H3K9ac | H4 | H4ac | all | notata | tata |
| PGD | FreeLB | 95.32 | 90.09 | 62.33 | 85.31 | 56.04 | **94.81** | **97.27** |
| | ADFAR | 93.53 | **98.33** | 60.58 | **95.96** | 83.52 | 40.20 | 89.77 |
| | AT | **96.32** | 93.99 | **63.34** | 85.31 | **98.47** | 49.07 | 76.80 |
| BertAttack | FreeLB | 0.00 | 0.00 | 0.00 | 0.00 | 0.00 | **16.33** | 0.00 |
| | ADFAR | 0.00 | 0.00 | 0.00 | 0.00 | 0.00 | 0.00 | 0.00 |
| | AT | 0.00 | 0.00 | 0.00 | 0.00 | 0.00 | 10.00 | 0.00 |
| TextFooler | FreeLB | 20.42 | 17.94 | 90.50 | 3.28 | 76.54 | **100.00** | 93.46 |
| | ADFAR | 63.24 | 15.85 | 88.98 | 81.68 | 65.48 | 92.86 | 89.93 |
| | AT | **99.64** | **45.01** | **92.80** | **85.59** | **100.00** | 27.44 | **93.97** |

| Attack | Defense | Transcription Factor Prediction (Human) | | | | | Core Promoter Detection | | |
|--------|---------|------|------|------|------|------|------|--------|--------|
| | | tf0 | tf1 | tf2 | tf3 | tf4 | all | notata | tata |
| PGD | FreeLB | **87.44** | 87.44 | 88.44 | 87.44 | **88.44** | **98.47** | 85.47 | **96.26** |
| | ADFAR | 83.42 | **99.50** | 76.38 | **95.48** | 87.44 | 68.94 | **98.61** | 90.77 |
| | AT | **87.44** | 87.44 | **91.46** | 87.44 | 79.40 | 96.10 | **98.61** | 83.30 |
| BertAttack | FreeLB | 2.13 | 0.00 | 2.04 | 0.00 | 0.00 | **6.82** | 0.00 | **1.92** |
| | ADFAR | 0.00 | 0.00 | 0.00 | 0.00 | 0.00 | 1.85 | 0.00 | 0.00 |
| | AT | **5.98** | 0.00 | **3.72** | 0.00 | 0.00 | 0.00 | 0.00 | 0.00 |
| TextFooler | FreeLB | 23.33 | 19.42 | 95.63 | **100.00** | 14.08 | 66.42 | 99.38 | 94.70 |
| | ADFAR | **100.00** | **100.00** | 89.68 | 87.00 | **100.00** | 93.89 | **100.00** | **100.00** |
| | AT | **100.00** | **100.00** | **100.00** | **100.00** | **100.00** | **100.00** | **100.00** | **100.00** |

| Attack | Defense | Transcription Factor Prediction (Mouse) | | | | |
|--------|---------|------|------|------|------|------|
| | | 0 | 1 | 2 | 3 | 4 |
| PGD | FreeLB | **94.47** | **98.59** | **75.38** | **83.76** | 89.81 |
| | ADFAR | 85.43 | 87.08 | 62.81 | 65.99 | 87.68 |
| | AT | **94.47** | 97.23 | 62.81 | 65.99 | **94.09** |
| BertAttack | FreeLB | 0.00 | 0.00 | 0.00 | 0.00 | 0.00 |
| | ADFAR | **37.04** | 0.00 | 0.00 | 0.00 | 0.00 |
| | AT | 1.23 | 0.00 | 0.00 | 0.00 | 0.00 |
| TextFooler | FreeLB | **100.00** | 19.69 | **100.00** | 94.94 | **80.78** |
| | ADFAR | **100.00** | **89.64** | **100.00** | **100.00** | 35.34 |
| | AT | **100.00** | 37.31 | **100.00** | **100.00** | 31.02 |

Table 23: **Performance Comparison of Adversarial Attack on Quantization Model.** This table reports the Attack Success Rate (ASR) of two adversarial attacks (TextFooler and BERTAttack) on quantized versions (Vanilla and Softmax$_1$) of DNABERT-2 and Nucleotide Transformer (NT) under W8A8 (8-bit weights and activations) quantization. All results are evaluated using the Attack Success Rate (ASR) metric.

| Attack | Model | Quant_Method | Epigenetic Marks Prediction | | | | | |
|---|---|---|---|---|---|---|---|---|
| | | | H3 | H3K14ac | H3K36me3 | H3K4me1 | H3K4me2 | H3K4me3 |
| TextFooler | DNABERT2 | Vanilla | **0.19** | **9.76** | **24.12** | **5.52** | 25.53 | **12.24** |
| | | Softmax$_1$ | 0.00 | 3.82 | 15.67 | 2.03 | **31.90** | 4.14 |
| | NT1 | Vanilla | 70.49 | **79.37** | **77.74** | **77.04** | **70.49** | **87.14** |
| | | Softmax$_1$ | **73.96** | 73.65 | 77.53 | 70.89 | 70.33 | 86.21 |
| BertAttack | DNABERT2 | Vanilla | **62.50** | 26.09 | **100.00** | 61.54 | 81.25 | **100.00** |
| | | Softmax$_1$ | **62.50** | **100.00** | 16.00 | **100.00** | **93.75** | 60.00 |
| | NT1 | Vanilla | **100.00** | **100.00** | **100.00** | **100.00** | **100.00** | **100.00** |
| | | Softmax$_1$ | 92.31 | **100.00** | **100.00** | **100.00** | **100.00** | 99.60 |

| Attack | Model | Quant_Method | Epigenetic Marks Prediction | | | | Promoter Detection (300bp) | | |
|---|---|---|---|---|---|---|---|---|---|
| | | | H3K79me3 | H3K9ac | H4 | H4ac | all | notata | tata |
| TextFooler | DNABERT2 | Vanilla | **4.30** | 0.00 | **11.48** | 1.30 | 27.58 | 17.05 | **30.29** |
| | | Softmax$_1$ | 3.96 | 0.00 | 4.19 | **1.48** | **28.21** | **22.44** | 29.55 |
| | NT1 | Vanilla | **71.49** | **73.37** | **56.52** | **72.17** | 59.54 | 54.59 | **58.15** |
| | | Softmax$_1$ | 68.89 | 67.25 | 55.12 | 71.90 | **68.42** | **63.40** | **58.15** |
| BertAttack | DNABERT2 | Vanilla | **100.00** | **100.00** | **57.14** | **99.78** | **98.08** | **96.43** | 72.56 |
| | | Softmax$_1$ | 84.62 | 87.50 | 0.00 | 96.15 | 66.11 | 70.00 | **100.00** |
| | NT1 | Vanilla | **100.00** | **100.00** | 91.67 | **100.00** | 98.25 | 93.75 | **100.00** |
| | | Softmax$_1$ | **100.00** | **100.00** | **99.27** | **100.00** | **100.00** | **97.83** | **100.00** |

| Attack | Model | Quant_Method | Transcription Factor Prediction (Human) | | | | | Core Promoter Detection | | |
|---|---|---|---|---|---|---|---|---|---|---|
| | | | tf0 | tf1 | tf2 | tf3 | tf4 | all | notata | tata |
| TextFooler | DNABERT2 | Vanilla | 1.17 | 0.00 | **14.07** | 38.34 | 0.20 | **63.88** | **67.90** | **61.33** |
| | | Softmax$_1$ | **13.45** | **5.61** | 11.49 | **38.67** | **4.48** | 62.36 | 61.12 | 48.87 |
| | NT1 | Vanilla | 57.41 | 51.93 | 67.28 | 74.05 | 53.26 | **66.18** | 63.73 | 42.81 |
| | | Softmax$_1$ | **69.22** | **65.50** | **71.97** | **77.68** | **69.39** | 59.52 | **68.14** | **49.06** |
| BertAttack | DNABERT2 | Vanilla | 0.00 | **11.11** | **63.64** | **100.00** | 16.67 | **97.83** | **64.29** | 89.02 |
| | | Softmax$_1$ | **2.91** | 2.91 | 26.58 | 80.00 | **32.47** | 96.71 | 36.79 | **98.55** |
| | NT1 | Vanilla | **100.00** | **100.00** | **100.00** | **100.00** | **100.00** | **100.00** | **100.00** | **100.00** |
| | | Softmax$_1$ | **100.00** | 96.43 | **100.00** | **100.00** | **100.00** | **100.00** | **100.00** | 99.60 |

Table 24: **Performance of Adversarial Attacks on HyenaDNA Trained with the GenoAdv Dataset.** This table compares the performance of HyenDNA trained with adversarial examples from the GenoAdv dataset. Three attack methods (BERTAttack, TextFooler, and PGD) are used to evaluate the models, with results reported in terms of Attack Success Rate (ASR). The best result is highlighted in bold, while the second-best result is underlined.

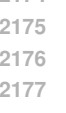
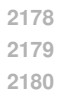
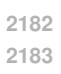

| | Epigenetic Marks Prediction | | | | | |
|---|---|---|---|---|---|---|
| Attack | H3 | H3K14ac | H3K36me3 | H3K4me1 | H3K4me2 | H3K4me3 |
| TextFooler | 1.01 | 5.41 | _83.24_ | _3.18_ | _17.86_ | 62.82 |
| PGD | _12.83_ | _19.29_ | 17.20 | 2.85 | 4.73 | 6.13 |
| BERT_Attack | **100.00** | **100.00** | **100.00** | **100.00** | **100.00** | **100.00** |

| | Epigenetic Marks Prediction | | | | Promoter Detection (300bp) | | |
|---|---|---|---|---|---|---|---|
| Attack | H3K79me3 | H3K9ac | H4 | H4ac | all | notata | tata |
| TextFooler | _26.27_ | _45.20_ | _33.53_ | _94.53_ | _44.20_ | _26.00_ | 1.05 |
| PGD | 12.56 | 16.90 | 20.16 | 7.71 | 21.13 | 10.06 | _20.27_ |
| BERT_Attack | **100.00** | **100.00** | **100.00** | **100.00** | **100.00** | **100.00** | **100.00** |

| | Transcription Factor Prediction (Human) | | | | | Core Promoter Detection | | |
|---|---|---|---|---|---|---|---|---|
| Attack | tf0 | tf1 | tf2 | tf3 | tf4 | all | notata | tata |
| TextFooler | 0.00 | 0.00 | 0.00 | 0.00 | 0.00 | 0.00 | 0.00 | 0.00 |
| PGD | _3.70_ | **40.00** | _19.15_ | _22.22_ | _19.15_ | _3.11_ | _13.83_ | _9.81_ |
| BERT_Attack | **70.37** | _15.00_ | **100.00** | **100.00** | **100.00** | **83.02** | **100.00** | **95.74** |

| | Transcription Factor Prediction (Mouse) | | | | |
|---|---|---|---|---|---|
| Attack | 0 | 1 | 2 | 3 | 4 |
| TextFooler | 0.00 | 0.00 | 0.00 | 0.00 | _23.94_ |
| PGD | _44.44_ | _7.06_ | _17.45_ | _15.79_ | 14.90 |
| BERT_Attack | **100.00** | **100.00** | **100.00** | **100.00** | **100.00** |

Table 25: **Performance of Adversarial Attacks on GenomeOcean Trained with the GenoAdv Dataset.** This table compares the performance of GenomeOcean trained with adversarial examples from the GenoAdv dataset. Three attack methods (BERTAttack, TextFooler, and PGD) are used to evaluate the models, with results reported in terms of Attack Success Rate (ASR). The best result is highlighted in bold, while the second-best result is underlined.

| | Epigenetic Marks Prediction | | | | | |
|---|---|---|---|---|---|---|
| Attack | H3 | H3K14ac | H3K36me3 | H3K4me1 | H3K4me2 | H3K4me3 |
| TextFooler | _62.66_ | **100.00** | **100.00** | **100.00** | **100.00** | **100.00** |
| PGD | 34.44 | 35.87 | 24.51 | 40.00 | 39.43 | 1.36 |
| BERT_Attack | **100.00** | _98.56_ | _97.65_ | **100.00** | **100.00** | **100.00** |

| | Epigenetic Marks Prediction | | | | Promoter Detection (300bp) | | |
|---|---|---|---|---|---|---|---|
| Attack | H3K79me3 | H3K9ac | H4 | H4ac | all | notata | tata |
| TextFooler | **100.00** | **100.00** | _63.89_ | **100.00** | **100.00** | **100.00** | 22.65 |
| PGD | 39.52 | 36.69 | 26.34 | 34.64 | 33.45 | 34.76 | _30.91_ |
| BERT_Attack | _95.70_ | **100.00** | **97.94** | _98.77_ | **100.00** | _96.45_ | **100.00** |

| | Transcription Factor Prediction (Human) | | | | | Core Promoter Detection | | |
|---|---|---|---|---|---|---|---|---|
| Attack | tf0 | tf1 | tf2 | tf3 | tf4 | all | notata | tata |
| TextFooler | **100.00** | **100.00** | **100.00** | **100.00** | **99.89** | _98.32_ | **100.00** | 22.71 |
| PGD | 34.18 | 12.68 | 35.80 | 19.15 | 35.65 | 44.22 | 40.89 | _39.07_ |
| BERT_Attack | _98.12_ | **100.00** | **100.00** | **100.00** | **100.00** | **98.84** | **100.00** | **100.00** |

| | Transcription Factor Prediction (Mouse) | | | | |
|---|---|---|---|---|---|
| Attack | 0 | 1 | 2 | 3 | 4 |
| TextFooler | 24.73 | _96.33_ | 13.58 | 8.88 | _80.71_ |
| PGD | _35.06_ | 30.33 | _34.42_ | _26.60_ | 25.45 |
| BERT_Attack | **100.00** | **100.00** | **98.96** | **100.00** | **100.00** |

Table 26: **Performance of Adversarial Attacks on DNABERT-2 Trained with the GenoAdv Dataset.** This table compares the performance of DNABERT-2 trained with adversarial examples from the GenoAdv dataset. Three attack methods (BERTAttack, TextFooler, and PGD) are used to evaluate the models, with results reported in terms of Attack Success Rate (ASR). The best result is highlighted in bold, while the second-best result is underlined.

| | Epigenetic Marks Prediction | | | | | |
|---|---|---|---|---|---|---|
| Attack | H3 | H3K14ac | H3K36me3 | H3K4me1 | H3K4me2 | H3K4me3 |
| TextFooler | _61.83_ | **100.00** | **100.00** | **100.00** | **100.00** | **100.00** |
| PGD | 39.53 | 24.67 | 34.53 | 36.71 | 35.61 | 34.79 |
| BERT_Attack | **87.67** | _85.36_ | **100.00** | _88.63_ | _88.13_ | **100.00** |

| | Epigenetic Marks Prediction | | | | Promoter Detection (300bp) | | |
|---|---|---|---|---|---|---|---|
| Attack | H3K79me3 | H3K9ac | H4 | H4ac | all | notata | tata |
| TextFooler | **99.88** | _69.87_ | _61.00_ | **100.00** | 56.26 | **100.00** | 24.27 |
| PGD | 41.24 | 29.06 | 26.35 | 37.59 | 38.23 | 45.11 | _44.93_ |
| BERT_Attack | _88.90_ | **100.00** | **87.10** | **100.00** | **100.00** | _88.99_ | **87.56** |

| | Transcription Factor Prediction (Human) | | | | | Core Promoter Detection | | |
|---|---|---|---|---|---|---|---|---|
| Attack | tf0 | tf1 | tf2 | tf3 | tf4 | all | notata | tata |
| TextFooler | **100.00** | _99.87_ | **100.00** | **100.00** | **99.21** | **100.00** | **100.00** | 23.39 |
| PGD | 30.12 | 25.33 | 24.39 | 2.22 | 28.09 | 36.36 | 22.71 | _36.89_ |
| BERT_Attack | _95.60_ | **100.00** | **100.00** | _97.78_ | _98.88_ | **100.00** | _98.80_ | **100.00** |

| | Transcription Factor Prediction (Mouse) | | | | |
|---|---|---|---|---|---|
| Attack | 0 | 1 | 2 | 3 | 4 |
| TextFooler | 28.54 | **98.28** | _12.77_ | 6.49 | _81.43_ |
| PGD | _35.81_ | 30.25 | 9.64 | _13.00_ | 34.63 |
| BERT_Attack | **100.00** | _87.94_ | **87.59** | **96.61** | **100.00** |

Table 27: **Performance of Adversarial Attacks on NT Trained with the GenoAdv Dataset.** This table compares the performance of Nucleotide Transformers (NT) trained with adversarial examples from the GenoAdv dataset. Three attack methods (BERTAttack, TextFooler, and PGD) are used to evaluate the models, with results reported in terms of Attack Success Rate (ASR). The best result is highlighted in bold, while the second-best result is underlined.

| | Epigenetic Marks Prediction | | | | | |
|---|---|---|---|---|---|---|
| Attack | H3 | H3K14ac | H3K36me3 | H3K4me1 | H3K4me2 | H3K4me3 |
| TextFooler | _56.41_ | _70.39_ | _77.72_ | _85.08_ | _77.87_ | _80.64_ |
| PGD | 28.57 | 23.43 | 21.88 | 29.53 | 21.67 | 22.90 |
| BERT_Attack | **100.00** | **100.00** | **100.00** | **100.00** | **100.00** | **100.00** |

| | Epigenetic Marks Prediction | | | | Promoter Detection (300bp) | | |
|---|---|---|---|---|---|---|---|
| Attack | H3K79me3 | H3K9ac | H4 | H4ac | all | notata | tata |
| TextFooler | _79.42_ | _69.67_ | _52.19_ | _66.39_ | _46.25_ | _64.64_ | _21.50_ |
| PGD | 17.64 | 26.87 | 7.49 | 19.89 | 19.39 | 7.97 | 7.83 |
| BERT_Attack | **100.00** | **100.00** | **100.00** | **100.00** | **100.00** | **100.00** | **100.00** |

| | Transcription Factor Prediction (Human) | | | | | Core Promoter Detection | | |
|---|---|---|---|---|---|---|---|---|
| Attack | tf0 | tf1 | tf2 | tf3 | tf4 | all | notata | tata |
| TextFooler | _58.31_ | _61.81_ | _46.13_ | _60.44_ | _67.96_ | _44.69_ | _67.92_ | _13.82_ |
| PGD | 28.57 | 24.15 | 21.57 | 25.48 | 10.11 | 23.01 | 25.96 | 13.01 |
| BERT_Attack | **100.00** | **85.37** | **100.00** | **97.85** | **98.88** | **100.00** | **100.00** | **100.00** |

| | Transcription Factor Prediction (Mouse) | | | | |
|---|---|---|---|---|---|
| Attack | 0 | 1 | 2 | 3 | 4 |
| TextFooler | 24.55 | _76.23_ | 10.08 | 8.26 | _66.19_ |
| PGD | _25.00_ | 21.96 | _10.71_ | _26.81_ | 26.46 |
| BERT_Attack | **100.00** | **100.00** | **100.00** | **100.00** | **100.00** |

Table 28: **Performance of Adversarial Attacks on NT2 Trained with the GenoAdv Dataset.** This table compares the performance of Nucleotide Transformers-2 (NT2) trained with adversarial examples from the GenoAdv dataset. Three attack methods (BERTAttack, TextFooler, and PGD) are used to evaluate the models, with results reported in terms of Attack Success Rate (ASR). The best result is highlighted in bold, while the second-best result is underlined.

| | Epigenetic Marks Prediction | | | | | |
|---|---|---|---|---|---|---|
| Attack | H3 | H3K14ac | H3K36me3 | H3K4me1 | H3K4me2 | H3K4me3 |
| TextFooler | 65.28 | **100.00** | **100.00** | **100.00** | **100.00** | **100.00** |
| PGD | 29.13 | 23.43 | 21.88 | 29.53 | 31.75 | 22.90 |
| BERT_Attack | **100.00** | **100.00** | 99.84 | **100.00** | 95.67 | **100.00** |

| | Epigenetic Marks Prediction | | | | Promoter Detection (300bp) | | |
|---|---|---|---|---|---|---|---|
| Attack | H3K79me3 | H3K9ac | H4 | H4ac | all | notata | tata |
| TextFooler | **100.00** | **100.00** | 63.67 | **100.00** | 53.67 | **100.00** | 24.35 |
| PGD | 24.51 | 26.87 | 28.29 | 22.67 | 29.39 | 2.19 | 13.01 |
| BERT_Attack | **100.00** | **100.00** | **91.56** | **100.00** | **100.00** | **100.00** | **100.00** |

| | Transcription Factor Prediction (Human) | | | | | Core Promoter Detection | | |
|---|---|---|---|---|---|---|---|---|
| Attack | tf0 | tf1 | tf2 | tf3 | tf4 | all | notata | tata |
| TextFooler | **100.00** | **100.00** | **100.00** | **100.00** | **100.00** | **100.00** | **100.00** | 24.50 |
| PGD | 22.17 | 21.76 | 26.96 | 23.33 | 26.32 | 45.80 | 28.48 | 28.69 |
| BERT_Attack | 99.81 | **100.00** | 98.91 | **100.00** | **100.00** | **100.00** | **100.00** | **100.00** |

| | Transcription Factor Prediction (Mouse) | | | | |
|---|---|---|---|---|---|
| Attack | 0 | 1 | 2 | 3 | 4 |
| TextFooler | 31.09 | **100.00** | 13.31 | 8.88 | 80.71 |
| PGD | 9.09 | 28.69 | 13.56 | 26.81 | 28.02 |
| BERT_Attack | **100.00** | 98.99 | **100.00** | **100.00** | **100.00** |

