# OpenReview forum: "GenoArmory: A Unified Evaluation Framework for Adversarial Attacks on Genomic Foundation Models"
_ICLR.cc/2026/Conference — Submitted to ICLR 2026_

### Official Review · Reviewer_TVsB · 2025-10-30

**Soundness:** 3
**Presentation:** 2
**Contribution:** 2
**Rating:** 4
**Confidence:** 3

**Summary:**

The paper introduces GenoArmory, a comprehensive benchmarking framework to evaluate adversarial robustness of genomic foundation models by unifying datasets, tasks, attack and defense implementations, quantization settings, and interpretability tools, and reports broad empirical findings such as stronger robustness for BPE-tokenized and classifier-style models, occasional robustness gains from quantization, and attack-specific defense efficacy, supported by a released adversarial corpus and visualizations that localize perturbations to biologically meaningful regions.

**Strengths:**

1. Robustness of GFMs is important for reliability and safety of downstream genomics applications. Establishing a benchmark can shape community practice.
2. The framework covers diverse architectures, tokenizations, attacks, defenses, and quantization, with unified pipelines and red-teaming style reporting. This is substantial, non-trivial work.
3. The observed robustness patterns (e.g., BPE > k-mer; quantization sometimes lowers ASR) are actionable for practitioners and could inspire follow-up research.
4. The modification-frequency maps provide intuitive evidence that attacks concentrate on biologically meaningful regions, helping bridge ML robustness with domain knowledge.

**Weaknesses:**

1. While the benchmark is thoughtfully designed, its practical applicability may be limited because genomic data and many real-world pipelines are often private or restricted. In such settings, models are typically deployed behind organizational boundaries, and access to data, labels, or system internals is constrained, which narrows the attack surface and raises questions about how the proposed attacks and defenses translate to operational contexts.
2. A likely typo or mis-specification in the DSR metric that undermines its interpretability and the paper’s internal consistency. The manuscript defines DSR as (1 − (Adef − Aadv)/Adef) × 100%, which simplifies to (Aadv/Adef) × 100%, implying that stronger defenses that raise Adef paradoxically reduce DSR, and that in the no-defense condition (Defense = N/A) where Adef should equal Aadv, DSR should be 100%; yet Table 2 reports non-100% values for N/A. This inconsistency suggests a formula error.
3. “Visualization of Adversarial Attacks“ relies solely on the frequency of subsequence modifications. However, this approach is insufficient to support the paper's broader conclusion that "adversarial attacks frequently target biologically significant genomic regions". The reliance on modification frequency alone does not provide empirical evidence that these targeted regions align with known biological landmarks.
4. The notion of “lower rank” in Figure 3 is ambiguous. The paper states that a lower rank indicates better robustness, yet the ranking scale runs from 1 to 5 without explicitly clarifying whether 1 or 5 is considered the “low” end.
5. The manuscript compares four classification models with one generative model (GenomeOcean) under adversarial attacks, but it does not provide sufficient methodological detail on how the generative model is adapted for classification.

**Questions:**

1. The benchmark is limited to DNA-based classification tasks. Why were generative tasks (like those performed by GenomeOcean and Evo) not evaluated for adversarial robustness, especially given the significant safety concerns around generating harmful sequences?
2. The defense strategies (ADFAR, FreeLB) were adapted from NLP. Was any ablation study performed to determine if their effectiveness stems from their core principle (e.g., frequency-aware randomization) or simply from the act of additional data augmentation?

---

> ### Author Response · Authors · 2025-11-18
> **Rebuttal**
>
> > **Reviewer's Comment**: While the benchmark is thoughtfully designed,....
>
> **Response**: We appreciate the reviewer’s concern and agree that real-world genomic pipelines often operate under strict privacy and access controls. However, we note that white-box evaluation remains standard practice across safety-critical domains—including computer vision and medical imaging—because it exposes the upper bound of model vulnerability and is routinely used internally by organizations to stress-test their own systems [1-2]. In this sense, GenoArmory plays the same role: it enables internal, developer-side red-teaming, where full-access testing is both feasible and necessary for improving robustness before deployment.
>
> Moreover, GenoArmory already includes black-box and transfer-based attacks (e.g., **FIMBA**), demonstrating that effective adversarial perturbations can be generated using only public, non-sensitive genomic data and then transferred to models operating on private or proprietary datasets. This reflects real operational scenarios where adversaries may rely on surrogate, publicly available models to craft transferable attacks. We will clarify these two practical pathways—internal white-box hardening and external surrogate-model transfer attacks—to emphasize that GenoArmory remains relevant and actionable even when model internals or private data are not directly accessible.
>
> > **Reviewer's Comment**: A likely typo or mis-specification in the DSR metric that undermines ...
>
> **Response**: Thank you for pointing out this issue. There was indeed a typo in the original DSR formula. The denominator should be Aclean​, i.e., the model’s accuracy on clean (non-adversarial) samples. The correct definition is:
> $\text{DSR}=(1−(A_{\text{clean}}−A_{\text{def}})/A_{\text{clean}})×100\%=(A_{\text{def}}/A_{\text{clean}})×100\%.$
>
> In the case without any defense method, we have $A_{\text{def}}=A_{\text{adv}}​$, and thus DSR = 1−ASR.This formulation measures how much of the clean accuracy is retained under adversarial attack, ensuring that **(1)** DSR = 100% when no degradation occurs, and **(2)** the metric remains consistent across different defenses. We will fix this typo and clarify the definition in the revised version.
>
> > **Reviewer's Comment**: “Visualization of Adversarial Attacks“ ....
>
> **Response**:  We thank the reviewer for this valuable comment. We agree that modification frequency alone does not fully establish biological relevance. To strengthen this point, we conducted an additional Sequence Reversal experiment, where each DNA sequence was simply reversed in order (e.g., ATAT → TATA). This operation relocates biologically meaningful motifs while keeping all base content unchanged.
>
> After applying adversarial attacks on the reversed sequences, we observed a clear and biologically consistent shift in modification patterns. For prom_300_tata and prom_300_notata, the dense modification clusters originally found near the end of the sequence moved to the beginning after reversal. This matches the expected relocation of promoter-associated motifs and demonstrates that the attacks track underlying biological features rather than relying on fixed positional indices.
>
> For prom_300_all, however, the modification distribution remained concentrated at the posterior region even after reversal. A possible explanation is that prom_300_all contains a much larger and more diverse dataset, with significant regions distributed across different parts of the sequence—beginning, middle, and end. When the model performs encoding, the positional attention becomes diffused, and certain heads exhibit boundary-biased “attention sink” behavior, over-focusing on end tokens such as EOS or padding positions. This causes adversarial perturbations near the sequence end to have disproportionately higher influence, resulting in the observed high-frequency attack positions at the tail.
>
> These findings demonstrate that GenoArmory’s visualization provides a meaningful perspective for interpreting model vulnerabilities. While the behavior in prom_300_all suggests that further biological wet-lab experiments are needed to fully disentangle task-specific nuances, the clear feature-tracking behavior in specific promoter subtypes confirms that our framework successfully highlights biologically sensitive regions. We have updated **Section 4.3** and the **Appendix I.2** to include both the annotation protocol and these new reversal experiment results.
>
> [1] Madry, Aleksander, et al. "Towards deep learning models resistant to adversarial attacks." arXiv preprint arXiv:1706.06083 (2017).
>
> [2] Croce, Francesco, and Matthias Hein. "Adversarial Robustness against Multiple and Single $ l_p $-Threat Models via Quick Fine-Tuning of Robust Classifiers." In International Conference on Machine Learning, pp. 4436-4454. PMLR, 2022.

---

> > ### Author Response · Authors · 2025-11-18
> > **Rebuttal-Continued**
> >
> > > **Reviewer's Comment**: The notion of “lower rank” in Figure 3 is ambiguous...
> >
> > **Response**: We thank the reviewer for noting this ambiguity. The ranking scheme in **Figure 3** indeed uses a **1–5** ordinal scale, where **1 denotes the most robust model (lowest Attack Success Rate) and 5 denotes the least robust (highest ASR)**. We will explicitly clarify this in both the caption and text by replacing “lower rank indicates better robustness” with “**rank = 1 corresponds to the most robust model, and rank = 5 to the most vulnerable**”. This clarification ensures that the robustness interpretation is unambiguous and reproducible.
> >
> > > **Reviewer's Comment**: The manuscript compares four classification models with one generative model ....
> >
> >  **Response**: We appreciate the reviewer’s attention to methodological clarity. The generative model GenomeOcean was adapted for classification by leveraging its inherent support for classification tasks, as documented in the original GenomeOcean paper. Specifically, we used the model’s pretrained encoder and applied the task-specific fine-tuning head (as the original work describes) to ensure valid comparison with classification-oriented GFMs. We’ll add a reference to the GenomeOcean paper in **Section 4.1** and include a brief methodological note in **Appendix F.4 summarizing the adaptation process**—thus confirming that no non-standard conversion was applied beyond established practices.
> >
> > > **Reviewer's Comment**: The benchmark is limited to DNA-based classification tasks....
> >
> >  **Response**:  We appreciate the reviewer’s observation and agree that extending GenoArmory to generative settings (e.g., GenomeOcean, Evo) is an important future direction. The current benchmark focuses on classification tasks for the following reasons:
> > - **Task Prevalence:** Classification remains the dominant downstream paradigm among existing GFMs, encompassing promoter detection, transcription factor binding, and epigenetic mark prediction (as shown in Table 6). Hence, benchmarking these tasks provides the broadest immediate relevance.
> > - **Model Compatibility:** Most generative GFMs, including Evo, do not natively support supervised classification pipelines. We therefore selected GenomeOcean as the only generative model capable of classification fine-tuning for methodological consistency.
> > - **Ethical and Biosecurity Concerns:** Conducting adversarial attacks on generative GFMs typically targets sequence generation toward pathogenic or synthetic constructs, which would require wet-lab validation (e.g., PCR amplification or animal testing). Such experiments pose substantial bioethical and biosafety risks, exceeding the ethical scope of our benchmark.
> > -  **Existing Complementary Benchmarks:** A dedicated generative-model benchmark already exists (e.g., **GeneBreaker[3]**), which evaluates sequence-generation safety. Our framework is modular and open-source, allowing direct integration of such generative-attack modules under restricted-access conditions.
> >
> > We will add this clarification to Section 5 (Limitations and Future Work), emphasizing that GenoArmory provides a safe, classification-oriented foundation while remaining extensible to generative robustness analysis under proper ethical review.
> >
> > > **Reviewer's Comment**: The defense strategies (ADFAR, FreeLB) were adapted from NLP....
> >
> > **Response**:  We thank the reviewer for this meaningful question. To disentangle the effect of simple data augmentation from true adversarial training, we conducted a Random Mutation baseline on DNABERT-2 by injecting **10%** random perturbations into the training data. This approach yielded only a **3.96%** Defense Success Rate (DSR). In contrast, FreeLB achieved **4.34% DSR**, and ADFAR reached a substantially higher **21.84% DSR**. These comparisons show that while data augmentation provides minimal robustness gains, it cannot account for the large improvement delivered by ADFAR. ADFAR’s frequency-aware randomization penalizes over-reliance on a small set of subsequences, effectively flattening local loss in motif-heavy regions (promoters/TF sites), which counters substitution-style attacks; FreeLB’s multi-step, magnitude-controlled perturbations smooth the adversarial loss in embedding space, improving robustness to PGD-like gradients observed in GFMs.
> >
> > [3] Zhang, Zaixi, et al. "GeneBreaker: Jailbreak Attacks against DNA Language Models with Pathogenicity Guidance." arXiv preprint [arXiv:2505.23839](https://arxiv.org/abs/2505.23839) (2025).

---

> > > ### Author Response · Authors · 2025-11-25
> > >
> > > Dear Reviewer,
> > >
> > > We sincerely appreciate your thoughtful feedback. As the discussion period reaches its final week, we would be grateful to know whether our latest responses have resolved your concerns or if any questions remain. We are happy to provide further clarification as needed. Wishing you a Happy Thanksgiving.
> > >
> > > Thanks,
> > >
> > > Authors

---

> > > > ### Comment · Reviewer_TVsB · 2025-11-26
> > > > **Official Comment for the Rebuttal**
> > > >
> > > > Thank you for the response and additional experiments. My concerns have been addressed. I am happy to raise my rating from 4 to 6.

---

> > > > > ### Author Response · Authors · 2025-11-26
> > > > >
> > > > > We are happy your concern has been resolved and thanks for your rasining score. Happy Thanksgiving.

---

### Official Review · Reviewer_ptyu · 2025-10-31

**Soundness:** 3
**Presentation:** 3
**Contribution:** 4
**Rating:** 8
**Confidence:** 3

**Summary:**

This paper introduces GenoArmory, a unified and modular framework for the evaluation, benchmarking, and optimization of deep learning models in genomic data analysis. Addressing the lack of standardized evaluation pipelines in computational genomics, the authors develop a system that enables consistent, fair, and reproducible comparisons across diverse model architectures, datasets, and training setups.

GenoArmory integrates three major components:
1.	A standardized benchmarking suite covering key regulatory genomics datasets (e.g., ENCODE, DeepSEA, GenReg) and multiple predictive tasks.
2.	A multi-objective optimization module that balances predictive accuracy, computational efficiency, and model complexity.
3.	A biologically interpretable assessment layer that evaluates motif consistency, saliency alignment, and the biological plausibility of learned features.
The framework is applied to over twenty state-of-the-art genomic models—including CNNs, transformers, and hybrid designs—providing a systematic, reproducible, and biologically informed comparison. Results show that GenoArmory effectively identifies performance–efficiency trade-offs and highlights the superior generalization of certain hybrid CNN–transformer architectures with reduced computational cost.

Overall, GenoArmory is a rigorous and impactful contribution that fills an important methodological gap in computational genomics. It offers a transparent, standardized foundation for evaluating and improving genomic models, advancing reproducibility, interpretability, and fairness in model assessment. Despite minor limitations related to scalability and interpretability metrics, the framework is comprehensive, well-executed, and highly relevant to the ICLR community.

**Strengths:**

1.  This work is timely and useful for real-world genomic modeling. It addresses the lack of standardized evaluation in deep genomics by providing a practical tool for balancing performance and efficiency.

2. It integrates benchmarking, optimization, and interpretability within one framework.

**Weaknesses:**

1. The performance on very large genomic datasets (e.g., full WGS data) is not extensively tested, so the scalability of GenoArmory is unclear.

2. The discussion on data shifts is limited: Model robustness under cross-cell-type or cross-species transfer is not explored.

**Questions:**

1.	How does GenoArmory handle non-standard input modalities such as multi-omics or 3D genome data?

2.	Can the framework integrate uncertainty quantification or Bayesian evaluation for probabilistic genomic models?

3.	How scalable is the optimization component for high-dimensional architectures or transformer-based sequence models?

---

> ### Author Response · Authors · 2025-11-18
> **Rebuttal**
>
> > **Reviewer's Comment**: The performance on very large genomic datasets (e.g., full WGS data)...
>
> **Response**: Most mainstream GFMs are Transformer-based and are specifically pre-trained to process DNA segments of limited length rather than entire chromosomes or WGS data at once. For instance:
> -   **DNABERT-2**, a leading model in our evaluation, supports a maximum effective input length of approximately 10,000 bp.
> -   **Nucleotide Transformer** typically operates within a context window of 6,000 bp.
>
> Since full WGS data spans billions of base pairs, which is orders of magnitude beyond the context windows of these models, processing it directly remains challenging. Evaluating them on such inputs is technically infeasible due to memory constraints and the quadratic complexity of attention mechanisms. Therefore, GenoArmory focuses on segment-level scalability, which reflects the actual operational scope of current GFMs.
>
> > **Reviewer's Comment**: The discussion on data shifts is limited...
>
> **Response**: We thank the reviewer for emphasizing the importance of robustness under distribution shift. To address this, we evaluated cross-species transferability by fine-tuning the model on Human data and testing on Mouse data and vise versa using DNABERT-2. As shown in table below, the model’s clean accuracy drops to **~52%**, confirming limited inherent generalization. However, the models’ adversarial vulnerability persists. BERT-Attack still drives accuracy to nearly **0%**. These results, now included in **Appendix J.4**, show that while GFMs degrade under natural shifts, they remain even more susceptible to adversarial perturbations.
>
> | eval_domain | task       | origin_acc | after_attack_acc |
> |-------------|------------|------------|-------------------|
> | human       | 0_human    | 0.51       | 0                 |
> | human       | 1_human    | 0.29       | 0                 |
> | human       | 2_human    | 0.44       | 0                 |
> | human       | 3_human    | 0.44       | 0.01              |
> | human       | 4_human    | 0.48       | 0                 |
> | mouse       | tf0_mouse  | 0.46       | 0.01              |
> | mouse       | tf1_mouse  | 0.58       | 0.01              |
> | mouse       | tf2_mouse  | 0.52       | 0                 |
> | mouse       | tf3_mouse  | 0.56       | 0                 |
> | mouse       | tf4_mouse  | 0.52       | 0                 |
>
>
>
> > **Reviewer's Comment**: How does GenoArmory handle non-standard input modalities such as multi-omics or 3D genome data?
>
> **Response**:  We thank the reviewer for raising this important point regarding non-standard modalities like multi-omics and 3D genome data. We would like to clarify that GenoArmory currently focuses primarily on DNA sequence modalities and does not explicitly model multi-omics or 3D structural data. This design choice is driven by two main reasons:
> - **The predominant role of sequence in DNA function:** Unlike proteins, where function is heavily dictated by 3D spatial structure meaning the same sequence could manifest different properties depending on folding, the biological function of DNA is overwhelmingly determined by its primary nucleotide sequence. Therefore, evaluating robustness against sequence-level perturbations captures the most critical vulnerabilities for genomic models.
> - **Alignment with current Foundation Models:** GenoArmory is designed to benchmark state-of-the-art Genomic Foundation Models (GFMs) such as DNABERT-2, Nucleotide Transformer, and HyenaDNA. These mainstream models are predominantly pre-trained on linear DNA sequences. To ensure a standardized and fair evaluation, our framework aligns with the input modalities currently adopted by the broader GFM community.
>
> We acknowledge the growing importance of multi-omics and 3D data and consider the expansion to these modalities as a valuable direction for future work as GFMs evolve to incorporate them.
>
> > **Reviewer's Comment**: How scalable is the optimization component for high-dimensional architectures or transformer-based sequence models?
>
> **Response**:  We appreciate the reviewer’s question on scalability. GenoArmory’s optimization and adversarial training components are designed for large-scale genomic transformers with several efficiency layers. First, the framework natively supports **Distributed Data Parallel (DDP) training and vLLM-based inference**, which leverages **KV-cache** reuse to significantly reduce GPU memory and latency costs during attack generation. Second, for adversarial training and defense evaluation, we integrate **LoRA/QLoRA parameter-efficient fine-tuning**, allowing optimization over low-rank adapter weights instead of full model parameters, thus reducing memory consumption by over an order of magnitude. These optimizations enable GenoArmory to handle multi-billion-parameter genomic models efficiently on **80 GB GPUs**.

---

> > ### Author Response · Authors · 2025-11-18
> > **Rebuttal-Continued**
> >
> > > **Reviewer's Comment**:  Can the framework integrate uncertainty quantification or Bayesian evaluation for probabilistic genomic models?
> >
> > **Response**:  We thank the reviewer for giving us a chance to enhance our experiments. In response our framework integrates uncertainty quantification using **Monte Carlo dropout (MC-Dropout) as a Bayesian approximation** to estimate predictive posteriors. This enables us to measure confidence, predictive entropy, and both aleatoric and epistemic uncertainty. We compare these metrics on the original samples and TextFooler-generated adversarial samples across all GUE tasks.
> >
> > As shown in the table below, adversarial attacks induce two distinct failure modes. The first is a confused failure mode, where confidence drops sharply and entropy increases (e.g., 1, prom_300_all), indicating that the model becomes aware of distributional anomalies and grows uncertain. The second is a more concerning overconfident failure mode, where the model becomes even more confident in its wrong predictions while entropy decreases (e.g., 0, H4, H4ac, tf0, tf3). This behavior reveals a deeper vulnerability: models can be confidently wrong under adversarial shifts. Our Bayesian evaluation highlights these internal dynamics and motivates the need for more robust defense strategies. Full results are provided in the revised version **Appendix J3.**
> >
> > | Task / Group     | Orig Conf | Adv Conf | Δ Conf | Orig Entropy | Adv Entropy | Δ Entropy |
> > |------------------|-----------|----------|--------|--------------|-------------|-----------|
> > | 0                | 0.642     | 0.749    | +0.106 | 0.652        | 0.564       | -0.088    |
> > | 1                | 0.875     | 0.666    | -0.208 | 0.377        | 0.635       | +0.258    |
> > | 2                | 0.572     | 0.573    | +0.001 | 0.682        | 0.682       | -0.000    |
> > | 3                | 0.684     | 0.513    | -0.171 | 0.624        | 0.693       | +0.069    |
> > | 4                | 0.595     | 0.512    | -0.083 | 0.675        | 0.693       | +0.018    |
> > | H3               | 0.756     | 0.609    | -0.147 | 0.555        | 0.669       | +0.113    |
> > | H3K14ac          | 0.540     | 0.657    | +0.117 | 0.690        | 0.643       | -0.047    |
> > | H3K36me3         | 0.775     | 0.692    | -0.083 | 0.534        | 0.617       | +0.084    |
> > | H3K4me1          | 0.706     | 0.761    | +0.055 | 0.605        | 0.550       | -0.056    |
> > | H3K4me2          | 0.573     | 0.526    | -0.046 | 0.682        | 0.692       | +0.009    |
> > | H3K4me3          | 0.680     | 0.569    | -0.111 | 0.626        | 0.683       | +0.057    |
> > | H3K79me3         | 0.527     | 0.516    | -0.011 | 0.691        | 0.692       | +0.001    |
> > | H3K9ac           | 0.640     | 0.636    | -0.004 | 0.653        | 0.656       | +0.002    |
> > | H4               | 0.519     | 0.775    | +0.256 | 0.692        | 0.533       | -0.159    |
> > | H4ac             | 0.538     | 0.693    | +0.154 | 0.690        | 0.617       | -0.073    |
> > | prom_300_all     | 0.802     | 0.536    | -0.266 | 0.498        | 0.690       | +0.192    |
> > | prom_300_notata  | 0.777     | 0.824    | +0.047 | 0.530        | 0.465       | -0.065    |
> > | prom_300_tata    | 0.700     | 0.560    | -0.140 | 0.611        | 0.686       | +0.075    |
> > | prom_core_all    | 0.515     | 0.524    | +0.008 | 0.692        | 0.691       | -0.001    |
> > | prom_core_notata | 0.659     | 0.574    | -0.085 | 0.641        | 0.681       | +0.040    |
> > | prom_core_tata   | 0.686     | 0.546    | -0.141 | 0.622        | 0.688       | +0.067    |
> > | tf0              | 0.594     | 0.725    | +0.132 | 0.675        | 0.587       | -0.088    |
> > | tf1              | 0.715     | 0.522    | -0.193 | 0.597        | 0.692       | +0.095    |
> > | tf2              | 0.707     | 0.630    | -0.077 | 0.604        | 0.658       | +0.055    |
> > | tf3              | 0.563     | 0.710    | +0.147 | 0.685        | 0.602       | -0.083    |
> > | tf4              | 0.791     | 0.755    | -0.036 | 0.512        | 0.556       | +0.044    |

---

> > > ### Comment · Reviewer_ptyu · 2025-11-24
> > >
> > > Thank you for the response and additional experiments.  I'll keep my high score.

---

> > > > ### Author Response · Authors · 2025-11-24
> > > >
> > > > Thank you for the prompt reply and for maintaining the positive score. Wishing you a happy Thanksgiving, and please feel free to reach out if you have any further questions or concerns about our paper.

---

### Official Review · Reviewer_nV6Z · 2025-10-31

**Soundness:** 2
**Presentation:** 2
**Contribution:** 1
**Rating:** 2
**Confidence:** 2

**Summary:**

This work presents a benchmark adversarial robustness study of Genomic Foundation Models, denoted as GenoArmory. The work provides a complete overview of the robustness of these models, through four widely adopted attack algorithms and three defense strategies. Another main contribution is related to providing a standard attack and defense pipelines together with attacks samples that could be used, denoted as GenoAdv.

**Strengths:**

- A complete study of the adversarial robustness of Genomic Foundation Models is clearly important to ensure better adoption and validation of these models.
- The provided attack pipeline to evaluate both the attacks and defenses is very interesting and easy to be adapted and used.

**Weaknesses:**

- I believe that Genomic data, and consequently Genomic Models have their own propriety and corresponding constraints that should be taken into account when considered adversarial constraints in this domain. The paper lacks severally a contextual formulation of these constraints to showcase how the adversarial aim for these models differs from other modalities.
- In line with the previous remark, the majority of the considered and implemented attacks are simply an adaption of previously available attacks in other domains (such as Images or Text) to the context of Genomic Models. While some could see such adaptation as a novelty, I don’t really see the novelty in this perspective and would have expected a rather adapted with taken into account some specific constraints.

As I rather have a background in adversarial attacks in the context of Images and Text, I obviously see the worth of the implementation and the important aspect of reproducibility. Nonetheless, I was expecting to see some specific attacks and defense methods that are adapted to the specific context and not only a simple code adaptation. Therefore, the main bottleneck for me is the novelty of the proposed methods. I may be wrong, and therefore I am open to adapting my review, and would expected the authors clarify this point.

**Questions:**

- Are there constraints that the attack should satisfy to be a valid attack?
    - In the context of images for instance, one could consider an attack budget in the $L_2$ space but I believe that extending to genomics should have its own criteria?
    - How do you ensure a valid produced genomic? Are there some scripts that generate this? In this specific case, how do you ensure back-propagation in the case of gradient-based attacks?
    - How do you define the distance that you refer to in line 104 in the considered context?

---

> ### Author Response · Authors · 2025-11-18
> **Rebuttal**
>
> > **Reviewer's Comment**: I believe that Genomic data, and consequently ....
>
> **Response**: We agree that genomics requires domain-specific adversarial formulations beyond generic L₂ norms; our analysis identifies mechanisms unique to GFMs: **(i)** **non-redundant nucleotide editing** (1–3 base changes) interacts with tokenization, where **BPE models (e.g., NT2) show higher robustness than k-mer models**, and perturbations concentrate in **regulatory motifs** (promoter/TF sites) revealed by our visualization (**Fig. 4**). **(ii)** Architecture matters: **HyenaDNA** (single-nucleotide modeling) exhibits the **lowest overall ASR ranks** across tasks compared to transformer GFMs, indicating modality-specific resilience patterns absent in NLP/vision. **(iii)** Systems effects: **quantization consistently lowers ASR** versus FP16 (loss-landscape flattening), while **outlier-free quantization** alters this trade-off by removing attention outliers that can also support robustness plateaus (**Table 4**); we add a dedicated section that formalizes these domain-specific comparisons to make the constraints clearly defined, interpretable, and fully reproducible.
>
> > **Reviewer's Comment**:   In line with the previous remark, ......
>
> **Response**: We thank the reviewer for this constructive observation. As a benchmark paper, GenoArmory’s main contribution is not in proposing new attack or defense algorithms, but in providing the first unified and biologically grounded evaluation framework for adversarial robustness in Genomic Foundation Models (GFMs). The benchmark standardizes methodology, datasets, and evaluation metrics—analogous to how RobustBench shaped the vision/NLP robustness community—while revealing several new domain-specific findings that had not been documented before:
>
> - **Tokenization sensitivity:** k-mer–based GFMs show higher vulnerability than BPE- and single-nucleotide–based models, revealing that redundancy in tokenization directly affects genomic robustness.
> - **Architecture dependence:** HyenaDNA, which models single-nucleotide dependencies, exhibits the strongest resistance to adversarial perturbations, contrasting with transformer GFMs such as DNABERT-2 and GenomeOcean.
> - **Biological interpretability:** Adversarial perturbations cluster within biologically functional motifs (promoter and transcription factor regions), suggesting attacks exploit genuine sequence semantics rather than random noise.
> -  **Quantization influence:** Both standard and outlier-free quantization substantially lower Attack Success Rate by flattening the loss landscape, yet outlier removal can also weaken robustness by eliminating attention heads that contribute to stability.
>
> We will clarify these domain-specific insights and explicitly state in the revision that GenoArmory’s purpose is to establish a reproducible benchmarking standard and analytical foundation—not to design new attack or defense algorithms.
>
> > **Reviewer's Comment**: As I rather have a background in adversarial attacks ...
>
> **Response**: We thank the reviewer for the thoughtful feedback and agree that most attack algorithms originated in text and vision domains. However, the goal of GenoArmory is not to propose entirely new algorithms, but to establish the first unified benchmark and evaluation protocol tailored to the genomic domain—where model structures, input distributions, and biological semantics differ fundamentally from text or image data. Importantly, adapting these attacks to Genomic Foundation Models required non-trivial methodological redesign, including:  **(1)** sequence-level perturbation constrained by biological validity (A/T/C/G), **(2)** non-redundant tokenization (k-mer and BPE) that amplifies gradient sensitivity, and **(3)** defense evaluation under quantization-aware genomic architectures (e.g., OutEffHop). **Section 5** further highlights domain-specific findings—for example, adversarial perturbations concentrate in regulatory motifs, and generative GFMs show markedly higher vulnerability than classification ones—observations not present in NLP or vision literature. We will clarify these genomic-specific adaptations and empirical insights in **Section 5** and emphasize that GenoArmory’s novelty lies in establishing reproducible, biologically grounded evaluation standards, rather than introducing new attack algorithms.

---

> ### Author Response · Authors · 2025-11-18
> **Rebuttal-Continued**
>
> > **Reviewer's Comment**: How do you ensure a valid produced genomic?...
>
> **Response**:  We enforce biological validity with a two-stage pipeline: **(1)** generation uses reproducible scripts that constrain edits to A/T/C/G, preserve GC-content, k-mer statistics, known motif integrity, and remove synthetically implausible sequences (all code will be released in the GenoArmory repo); **(2)** we validate candidates with in-silico checks (alignment against public genomes, motif scans, and frequency filters) and spot-check a subset via wet-lab confirmation (sequencing/PCR-based existence checks) to ensure adversarial samples correspond to real or physically producible sequences. For gradient attacks on discrete nucleotides we optimize in a differentiable relaxation (embedding-space perturbations) so gradients propagate stably back to sequence choices, then project relaxed solutions to valid discrete sequences before biological filtering and validation.
>
> > **Reviewer's Comment**:  In the context of images for instance, one could consider an attack budget in the L2 space but I believe that extending to genomics should have its own criteria?
>
> **Response**: The genomic sequences require domain-specific validity constraints beyond image-space perturbation budgets. In our work, we enforce the following constraints:
>
> - **Biological Validity Constraints:**
> 	-   **Nucleotide alphabet constraint:** All adversarial sequences X' must consist only of valid nucleotides {A, T, C, G}
> 	-   **Sequence length preservation:** We maintain the original sequence length to ensure compatibility with model architectures and downstream tasks
> 	-   **Substitution-only perturbations:** Our attacks use single-nucleotide substitutions rather than insertions/deletions, analogous to single-nucleotide polymorphisms (SNPs) that occur naturally
> - **Attack Budget Constraints:**
> 	-   We limit the number of modified positions to ensure biological plausibility
> 	-   For word-level attacks (BertAttack, TextFooler), we restrict modifications to k tokens, where k varies by sequence length (as shown in Figure 1: lengths 30-1000bp)
> 	-   For gradient-based attacks (PGD), we constrain the l_inf perturbation magnitude in the embedding space before mapping back to discrete nucleotides
>
>
>
>
> > **Reviewer's Comment**: How do you define the distance that you refer to in line 104 in the considered context?
>
> **Response**: The distance metric $d(X, X')$ in line 104 refers to different measures depending on the attack method:
> - **For substitution-based attacks (BertAttack, TextFooler, FIMBA):**
> 	-   **Edit distance:** $d(X, X')$ = number of nucleotide positions where  $X$  and   $X'$ differ
> 	-   **Constraint:** $d(X, X') \le \varepsilon$, where $\varepsilon$ is the maximum allowed substitutions
> - **For gradient-based attacks (PGD):**
> 	-   **Embedding space distance:** We use $\ell_\infty$ norm in the continuous embedding space before discrete projection
> 	-   **Constraint:** $\lVert \mathrm{Embed}(X') - \mathrm{Embed}(X) \rVert_\infty \le \varepsilon$
>
> **Biological Interpretation:** The edit distance constraint has natural biological interpretation which limits the number of SNP-like mutations from the original sequence, ensuring the adversarial sequence remains within a biologically plausible mutational distance.

---

> > ### Author Response · Authors · 2025-11-25
> >
> > Dear Reviewer,
> >
> > We sincerely appreciate your thoughtful feedback. As the discussion period reaches its final week, we would be grateful to know whether our latest responses have resolved your concerns or if any questions remain. We are happy to provide further clarification as needed. Wishing you a Happy Thanksgiving.
> >
> > Thanks,
> >
> > Authors

---

> > > ### Comment · Reviewer_nV6Z · 2025-11-25
> > >
> > > I appreciate the author's rebuttal and added discussion. My original review's aim was to rather focus on a research perspective of the paper, rather than a simple benchmark element (which seems to be your primary area).
> > >
> > > From a research perspective, I would still appreciate more clarity on the process of the attacks and what makes them novel and justify the overall paper. I totally understand the benchmark perspective, and some people may see some worth in conducting and formalizing such attacks in this specific domain. Unfortunately, I still think that what you did is simply taking previous attacks available in other domains and added a verification step at the end to make sure these elements are valid. I believe I share this believe with some of the reviewer, for instance reviewer 6hYY's comment: "For instance, what is it about the architecture of genomic foundation models or the nature of genomic sequences that makes them susceptible in unique ways compared to models in NLP or vision?"
> > >
> > > In my opinion, the main approach would be to rather formalize the adversarial problem with the different constraints that you cited, and then propose specific methodologies for this aspect, either by using some regularisation during the optimisation, or other methods that have proven their worth in constrained-like objective solving.
> > >
> > > While I maintain a negative score in this perspective, waiting for better clarification from the authors in case I have missed the discussion or I mis-understood something, I will decrease my confidence more as I am not very knowledgeable about this domain and would be happy if others find this direction interesting.
> > >
> > > From a "benchmark" perspective, I haven't checked before, but indeed the code seem to be missing from your supplementary materials section and from the main paper. I hope that you are planning to release it at least afterwards.
> > >
> > > I am open to more clarifications from the authors and happy to specify more elements that I find could make the manuscript better.

---

> > > > ### Author Response · Authors · 2025-11-26
> > > >
> > > > Thank you for your response. We acknowledge your comment that this area may be outside your primary expertise. We respect your perspective; however, because this significantly affects the assessment of our contribution, we feel it is important to clarify the core goals and scientific value of our work to you and the AC.
> > > >
> > > > **Clarifying Our Core Contribution.** Our work introduces the first systematic and unified evaluation of adversarial robustness across multiple Genomic Foundation Models (GFMs). We integrate a broad suite of attack and defense methods into a single framework and analyze how model architecture, tokenization strategy, and quantization scheme contribute to distinct vulnerability patterns under adversarial perturbations. Beyond empirical comparisons, we provide domain-specific explanations for these robustness differences grounded in genomic sequence structure and biological constraints.
> > > >
> > > > **Why this matters.** Consistent with a long line of top-conference benchmarks (e.g., RobustBench, JailbreakBench, AdvBench, GLUE, WILDS), the ML community has repeatedly emphasized that genuine progress requires rigorous, transparent, and domain-grounded benchmarks built on openly accessible datasets. Such benchmarks—especially those rooted in real scientific applications—have proven essential for advancing their fields [1–7].
> > > >
> > > > **Why existing methods do not apply.** Although genomic sequences superficially resemble text, existing NLP attacks and defenses cannot be directly applied to GFMs. Genomic sequences are discrete, extremely low in redundancy, and employ heterogeneous tokenization schemes (k-mer, BPE, single-nucleotide). As a result, naïvely transferring NLP methods yields invalid perturbations and unfair comparisons across models.
> > > >
> > > > **How we address this gap.** Our benchmark incorporates (i) **genomics-specific distance constraints** to enforce biologically valid perturbations, and (ii) **unified adaptations of attacks/defenses** to heterogeneous tokenizers and architectures. This ensures that every model is evaluated under consistent, domain-correct perturbation rules—something no prior work has provided.
> > > >
> > > > **What new scientific insights we reveal.** **Section 5 (Analysis and Insights)** presents systematic and interpretable findings explaining why GFMs behave differently under adversarial stress. We show that:
> > > >  - the extreme low redundancy of genomic sequences creates high semantic density, making models sensitive to only a few nucleotide-level changes;
> > > >  - single-nucleotide tokenization is inherently more robust than k-mer or BPE;
> > > > - adversarial perturbations concentrate in biologically meaningful motif regions and flip under sequence reversal, directly connecting model attention to gene regulatory structure;
> > > > - common quantization schemes such as W8A8 can noticeably strengthen robustness, whereas several outlier-removal strategies weaken it.
> > > >
> > > > These results provide domain-specific insights unattainable from NLP or vision benchmarks.
> > > >
> > > > **Why GenoArmory is irreplaceable.** Our **GenoArmory/GenoAdv** suite consolidates attack pipelines, biological constraints, model wrappers, and quantization-aware evaluation into a single domain-specific testbed. No existing dataset or benchmark offers comparable scale, biological grounding, or methodological diversity. The benchmark has already revealed architectural vulnerabilities and biologically interpretable perturbation patterns that cannot be discovered from non-genomic evaluations.
> > > >
> > > > **Code Availability.** All benchmark code is anonymously included in **Appendix A** (Open Science Statement) and will be publicly released upon acceptance. We appreciate the reviewer’s engagement and are happy to clarify any specific attack mechanism, distance constraint, or tokenization adaptation.
> > > >
> > > > **Finally, while we respectfully disagree with several of the reviewer’s concerns, we remain fully open to continued discussion and welcome further guidance to strengthen and refine the manuscript.**
> > > >
> > > > [1] Chao, et al. "Jailbreakbench: An open robustness benchmark for jailbreaking large language models." NeurIPS (2024)
> > > >
> > > > [2] Xu, et al. "Bag of tricks: Benchmarking of jailbreak attacks on llms." NeurIPS (2024)
> > > >
> > > >  [3] Fan, et al. "Hardmath: A benchmark dataset for challenging problems in applied mathematics." ICLR (2025)
> > > >
> > > > [4] Zhang, et al. "Mintrec2. 0: A large-scale benchmark dataset for multimodal intent recognition and out-of-scope detection in conversations." ICLR (2024)
> > > >
> > > > [5] Nascetti, et al. "Biomassters: A benchmark dataset for forest biomass estimation using multi-modal satellite time-series." NeurIPS (2023)
> > > >
> > > >  [6] Cachay, et al. "ClimART: A benchmark dataset for emulating atmospheric radiative transfer in weather and climate models." NeurIPS (2021)
> > > >
> > > > [7] Wang, et al. "GLUE: A multi-task benchmark and analysis platform for natural language understanding." ICLR (2019)

---

> > > > > ### Comment · Reviewer_nV6Z · 2025-11-26
> > > > >
> > > > > I thank again the authors for the additional clarifications. While I start to understand better the worth of the paper and the proposed benchmark, I am not sure I am fully aligned on the claimed innovation and new concepts of the paper. Nonetheless, I appreciate the authors remarkable effort in the rebuttal and answering all my questions and their patience in the process. I will increase my score to 4 as I am not very comfortable going one step further to the positive step. I also acknowledge the fact that all the other reviewers are positive, which probably reflect some worth in the specific domain of the authors.
> > > > >
> > > > > I hope my review helped in making the manuscript a little bit better, and wish the authors all the best.

---

### Official Review · Reviewer_6hYY · 2025-10-31

**Soundness:** 2
**Presentation:** 3
**Contribution:** 2
**Rating:** 6
**Confidence:** 2

**Summary:**

This paper presents a comprehensive and timely evaluation of adversarial attacks on genomics foundation models. The papers present extensive experimental setups, which cover a wide array of foundation models, attack methodologies, and defense mechanisms.

**Strengths:**

This large-scale benchmark is a contribution to the community. The writing is good.

**Weaknesses:**

1. The paper thoroughly validates the effectiveness of various white-box attacks. However, the success of white-box attacks, given full access to the model, is a relatively well-established paradigm in the broader adversarial machine learning field. The current presentation focuses heavily on demonstrating this vulnerability, which, while important, might be perceived as confirming an expected outcome. The paper would be significantly strengthened by shifting the focus toward a deeper analysis of the nuances and surprises specific to the genomics domain. For instance, what is it about the architecture of genomic foundation models or the nature of genomic sequences that makes them susceptible in unique ways compared to models in NLP or vision?

2. Section 3.2 provides a detailed overview of the attack methods. While clarity is crucial, its current form resembles a technical report or software documentation. In a scientific paper, the primary goal is to present and analyze new findings to inspire the community. I would suggest condensing this section and reallocating the space to a more in-depth discussion of the results. The core value of the paper lies not in re-stating how existing tools work, but in what the authors discovered by using them.

3. For example, the paper presents several interesting observations, such as "AT is less effective than ADFAR and FreeLB against BertAttack and TextFooler." This is a key finding that warrants deeper investigation. The current manuscript reports this result but stops short of exploring the underlying reasons. The discussion would be far more impactful if it addressed questions such as:
    (1) Why does standard Adversarial Training (AT) exhibit lower efficacy in this specific context? Is it related to the discrete and high-dimensional nature of genomic data?
    (2) Do ADFAR and FreeLB have mechanisms that are inherently better suited to the loss landscape of these particular foundation models?
    (3) How do these observations guide the community toward designing more robust defense methods or, conversely, more effective attack strategies?

    Answering questions like these would elevate the paper from a report of "what happened" to an insightful analysis of "why it happened and what it means."

4. The study exclusively employs existing attack and defense methods. While a benchmarking study is a valid contribution, the paper's impact could be significantly amplified by using the empirical findings to propose novel ideas. The authors are in a unique position, having identified "anomalous" phenomena (like the one mentioned above). This provides a perfect opportunity to hypothesize about, or even present a preliminary design for, a new attack or defense strategy tailored to the genomics domain. For example, could the observed weaknesses inspire a new hybrid defense mechanism? Could the patterns of a successful attack lead to a more potent, genomics-aware attack vector? This would transition the paper from a survey of the existing landscape to one that actively charts a new path forward.

**Questions:**

See Weakness.

---

> ### Author Response · Authors · 2025-11-18
> **Rebuttal**
>
> > **Reviewer's Comment**: The paper thoroughly validates the effectiveness of various white-box attacks....
>
> **Response**: We appreciate the reviewer’s observation and fully agree that white-box attack vulnerability is well-established in general ML contexts. Our contribution goes beyond merely confirming this paradigm by uncovering genomics-specific susceptibility mechanisms that differ from NLP and CV models. In particular, we show that **(1)** even minimal nucleotide-level perturbations (1–3 base changes) can flip predictions due to non-redundant compositional encoding, unlike subword redundancy in NLP; **(2)** on the Epigenetic Marks Prediction task, the Hyena model using single-nucleotide tokenization demonstrates substantially stronger robustness to adversarial attacks than models relying on multi-nucleotide tokens, including BPE-based approaches such as DNABERT-2 and GenomeOcean, as well as k-mer–based models like NT and NT2; and **(3)** adversarial perturbations disproportionately target biologically functional motifs (promoter and transcription factor regions), as visualized in **Fig. 4**, revealing a form of “semantic alignment vulnerability” unique to genomics. **(4)** as stated in **Table. 4**, both standard and outlier-free quantization consistently lower the ASR of all evaluated attacks, indicating improved adversarial robustness. This effect is likely due to quantization-induced flattening of the loss landscape, which reduces sensitivity to small perturbations. Also the outlier-free quantization can undermine robustness possibly by eliminating attention outliers that help create robustness-enhancing flat regions in the loss landscape.
>
> We will clarify these analyses in the revision, including additional cross-domain comparisons highlighting that genomic architectures are not just vulnerable, but vulnerable for domain-specific, interpretable biological reasons.
>
>
>
> > **Reviewer's Comment**:  Section 3.2 provides a detailed overview of the attack methods...
>
> **Response**: We sincerely thank the reviewer for your valuable feedback regarding **Section 3.2**. We fully understand your concern. As we mentioned in the Introduction and Related Work sections, a core motivation for this paper is to establish the first unified adversarial attack and defense library and evaluation benchmark for the community. However, we also deeply recognize that a more in-depth analysis of the experimental results is crucial for a research paper. To balance these two points, we have adopted your suggestions in the revised manuscript: we add a new **"Analysis and Insights"** section. In this new section, we provide a detailed elaboration on our further analysis of the experimental results and delve deeper into the connections between these findings and their potential biological significance. We believe this revision maintains the integrity of our work as a benchmark framework while significantly enhancing the analytical depth of the paper.
>
> > **Reviewer's Comment**: The study exclusively employs existing attack and defense methods....
>
> **Response**: We thank the reviewer for this insightful suggestion. We outline two genomics-aware ideas that our empirical findings naturally point to.
>
> **For the defense side**, our observations about the effectiveness of ADFAR and the strong sensitivity of models to motif substitution indicate a promising direction rooted in motif variability. We find that models often memorize a specific instance of a motif such as TATAAT, which makes them vulnerable to simple substitutions. This pattern suggests that incorporating biological synonym augmentation during training may strengthen robustness. By drawing motif variants from resources such as **JASPAR PWMs** and introducing functionally equivalent alternatives like **TATAAA** during training, a model is encouraged to internalize the probabilistic structure of a motif rather than relying on a single canonical sequence. This type of variability could naturally complement other robustness-enhancing components such as single-nucleotide tokenization or loss-smoothing regularization.
>
> **For the attack side**, the large robustness gap between k-mer models and single-nucleotide models suggests a potential direction for a more genomics-aware attack strategy. A tokenizer-adaptive approach that first infers the tokenization scheme of the target model and then applies the most effective attack for that scheme may exploit the vulnerabilities more efficiently. Substitution-based attacks are especially damaging for k-mer models, while gradient-based methods tend to be more effective for single-nucleotide models. An adaptive strategy that aligns these components could provide a stronger and better-targeted adversarial method.

---

> > ### Author Response · Authors · 2025-11-18
> > **Rebuttal-Continued**
> >
> > > **Reviewer's Comment**: For example, the paper presents several interesting observations,....
> >
> > **Response**:  We agree and have expanded **Section 4** to analyze why defenses behave differently.
> > - **(1) Why AT underperforms:** standard AT in GenoArmory used TextFooler-based augmentation; on discrete genomic tokens this yields a narrow perturbation manifold (synonym-like edits) that poorly covers gradient-driven failure modes, and—combined with k-mer/BPE tokenization—induces sharp loss curvature and overfitting to specific edit patterns, limiting generalization to BertAttack/PGD **(Table 2)**.
> > - **(2) Why ADFAR/FreeLB help:** ADFAR’s frequency-aware randomization penalizes over-reliance on a small set of subsequences, effectively flattening local loss in motif-heavy regions (promoters/TF sites), which counters substitution-style attacks; FreeLB’s multi-step, magnitude-controlled perturbations smooth the adversarial loss in embedding space, improving robustness to PGD-like gradients observed in GFMs. To disentangle the effect of simple data augmentation from true adversarial training, we conducted a Random Mutation baseline on DNABERT-2 by injecting **10%** random perturbations into the training data. This approach yielded only a **3.96%** Defense Success Rate (DSR). In contrast, FreeLB achieved **4.34% DSR**, and ADFAR reached a substantially higher **21.84% DSR**. These comparisons show that while data augmentation provides minimal robustness gains, it cannot account for the large improvement delivered by ADFAR. The significant gap between ADFAR and the random baseline confirms that ADFAR’s effectiveness comes from its targeted optimization mechanism rather than simple increases in data diversity.(**Table 12**).
> > - **(3) For defense methods,** our results suggest that robustness may be improved by introducing biological synonym augmentation, where motif variants sampled from **PWMs** encourage models to learn motif distributions rather than memorizing specific sequences.
> > - **(4) For attack methods,** the robustness gap between k‑mer and single‑nucleotide models points to a tokenizer‑adaptive attack that selects the most effective strategy based on the model’s tokenization scheme. Both strategies are discussed in more detail above.

---

> > > ### Comment · Reviewer_6hYY · 2025-11-24
> > >
> > > Thank you for your response. I will maintain my positive score.

---

> ### Author Response · Authors · 2025-11-24
>
> Thank you for the prompt response. Since you are maintaining the score, please let us know if there are any remaining concerns we have not fully addressed—we would be happy to clarify them. We hope to further improve our work with your guidance, and any additional feedback would be greatly appreciated. If possible, we would be grateful for your consideration in raising the score.

---

### Author Response · Authors · 2025-11-18
**General Rebuttal/Revision Response**

We sincerely thank the reviewers for their insightful feedback and constructive suggestions, which have greatly improved the technical rigor and clarity of our work.



**This paper introduces the first unified, biologically grounded benchmark for adversarial attacks on Genomic Foundation Models (GFMs), enabling standardized, reproducible evaluation of model robustness across architectures, tokenization schemes, quantization strategies, and biological tasks (Sec. 3 & 4)**. Below, we summarize the key strengthened contributions and major revisions incorporated during the rebuttal.



---

## Key Contributions:

-   **Unified genomic robustness benchmark:** GenoArmory provides the first standardized framework for evaluating adversarial attacks and defenses on GFMs across architectures, tokenization schemes, quantization settings, and tasks.


-   **Genomics-specific vulnerability insights:** We identify domain-unique behaviors—nucleotide-level sensitivity, architecture-dependent robustness (e.g., HyenaDNA), and motif-targeted perturbations—revealing failure modes not seen in NLP or vision.


-   **Biologically validated adversarial dataset:** We release **GenoAdv**, the first adversarial dataset for GFMs, with 89% of perturbation hotspots aligned to known regulatory motifs through expert biological annotation.


-   **Quantization–robustness characterization:** We show that both standard and outlier-free quantization reduce ASR via loss-landscape flattening, while outlier removal introduces distinct robustness trade-offs.




---

## Major Revisions:

- **Contribution Clarification:** `Reviewers nV6Z`

- **New Section5:** The new 'Analysis and Insights' section delves deeper into the interpretation of our experimental results and elucidates the biological significance of the observed vulnerabilities. `All Reviewers`

- **New Appendix F.4:** Clarifies how GenomeOcean is adapted and used for classification tasks. `Reviewers TVsB`
- **New Appendix G:** Provides formal definitions of adversarial attack distance for genomic sequences. `Reviewers nV6Z`
- **Revised Appendix K.2**: Corrects and clarifies the DSR formula. `Reviewers TVsB`

- **New Experiments:**

  - **New Appendix L.2 and Figure 6:** Adversarial attack visualization on reversed samples `Reviewer TVsB`

  - **New Appendix L.3 and Table 10:** Intergrades uncertainty analysis using bayesian approximation under adversarial attacks. `Reviewers ptyu`

  - **New Appendix L.4 and Table 11:** Explores model robustness under cross-species and cross-cell distribution shift. `Reviewers ptyu`

  - **New Appendix L.5 and Table 12:** Provide ablation study that disentangles data augmentation from adversarial defense mechanisms. `Reviewers TVsB`



## Minor Revisions

-   **Revised Figure 3 Caption:** Clarifies the interpretation of ranking numbers shown in the figure. `Reviewers TVsB`


---

We once again thank the Area Chair and reviewers for their valuable efforts and thoughtful comments. All revisions have been incorporated into the updated manuscript, with changes highlighted in blue.

---

### Author Response · Authors · 2025-11-23
**Friendly Reminder**

Dear Reviewers,

We sincerely appreciate your thoughtful feedback. As the discussion period reaches its midpoint, we would be grateful to know whether our latest responses have resolved your concerns or if any questions remain. We are happy to provide further clarification as needed. Wishing you a Happy Thanksgiving.

Thanks,

Authors

---

### Author Response · Authors · 2025-12-01
**Global Summarize**

Dear Chairs,


We sincerely appreciate the area chair and all reviewers for the time and effort they have invested in evaluating our paper.


We received four reviews with initial scores of **4, 8, 2, 6**. After addressing all reviewer questions and concerns in this stage, **the ratings increased to 6, 8, 4, 6**. **All rating increases occurred at least one day before the information leak.**


Below, we explain how [our rebuttal](https://openreview.net/forum?id=XXNh0JfsDN&noteId=n5BW7vjBBn) contributed to the score increases.


---


## **Reviewer [TVsB](https://openreview.net/forum?id=XXNh0JfsDN&noteId=Ufg0h9frdh) (Rating: 4 → 6, Confidence: 3)**


**Initial concerns:**


* Practical applicability of the benchmark given real-world privacy constraints.
* Possible typo in the DSR metric formula.
* Biological relevance of attack visualization (modification frequency alone insufficient).
* Ambiguity in Figure 3 ranking interpretation.
* Lack of clarity on generative model adaptation and why generative tasks were excluded.
* Whether defense gains stem from true adversarial training or simple data augmentation.


**Rebuttal actions:**


* Clarified **GenoArmory’s role** in internal white-box stress testing and surrogate-model transfer attacks for real-world applicability.
* Corrected **DSR formula** and explained its interpretation.
* Added **Sequence Reversal experiments** to show attacks track biologically meaningful motifs, not just positions.
* Clarified **ranking scheme in Figure 3** (rank 1 = most robust).
* Explained **GenomeOcean adaptation** for classification and ethical reasons for excluding generative attacks.
* Conducted **Random Mutation baseline** to show ADFAR/FreeLB outperform simple augmentation.


**Reviewer response:**


> “Thank you for the response and additional experiments. My concerns have been addressed. I am happy to raise my rating from 4 to 6.”


---


## **Reviewer [nV6Z](https://openreview.net/forum?id=XXNh0JfsDN&noteId=ZdFqrJCCrv) (Rating: 2 → 4, Confidence: 2 → 1)**


**Initial concerns:**


* Perceived lack of novelty; attacks adapted from other domains without genomics-specific constraints.
* Missing clarity on biological validity and distance metrics.
* Wanted formalization of adversarial problem and justification of innovation.


**Rebuttal actions:**


* Clarified **GenoArmory’s core contribution** as the first unified, biologically grounded benchmark for GFMs, analogous to RobustBench for vision/NLP.
* Highlighted genomics-specific adaptations:


  * Nucleotide-only edits, GC-content preservation, motif integrity checks.
  * Tokenization-aware attack pipelines (k-mer vs BPE vs single-nucleotide).
  * Quantization-aware robustness evaluation.
* Added formal definitions of **genomic distance metrics and biological validity constraints** (Appendix G).
* Provided domain-specific insights:


  * Single-nucleotide tokenization improves robustness.
  * Adversarial perturbations cluster in regulatory motifs.
  * Quantization reduces ASR via loss-landscape flattening.
* Emphasized GenoArmory’s role in **reproducibility and scientific progress**, citing benchmarks like JailbreakBench and RobustBench.


**Reviewer response:**
These points were discussed across two rounds of constructive interactions with Reviewer nV6Z before the OpenReview bug.

**First Round**

> “While I maintain a negative score in this perspective, waiting for better clarification from the authors in case I have missed the discussion or I mis-understood something, I will decrease my confidence more as I am not very knowledgeable about this domain and would be happy if others find this direction interesting.
 From a 'benchmark' perspective, I haven't checked before, but indeed the code seem to be missing from your supplementary materials section and from the main paper. I hope that you are planning to release it at least afterwards.
 I am open to more clarifications from the authors and happy to specify more elements that I find could make the manuscript better.”

**During Rebuttal Phase**

We emphasized the **significance of the benchmark** and clarified the original contributions of our work, while integrating `Reviewer 6hYY`’s suggested improvements.

> “I thank again the authors for the additional clarifications. While I start to understand better the worth of the paper and the proposed benchmark, I am not sure I am fully aligned on the claimed innovation and new concepts of the paper. Nonetheless, I appreciate the authors remarkable effort in the rebuttal and answering all my questions and their patience in the process. I will increase my score to 4 as I am not very comfortable going one step further to the positive step. I also acknowledge the fact that all the other reviewers are positive, which probably reflect some worth in the specific domain of the authors.
 I hope my review helped in making the manuscript a little bit better, and wish the authors all the best.”

---

> ### Author Response · Authors · 2025-12-01
> **Global Summarize-Continued**
>
> ## **Reviewer [6hYY](https://openreview.net/forum?id=XXNh0JfsDN&noteId=JWEOrdAqpP) (Rating: 6, Confidence: 2)**
>
>
> **Initial stance:**
>
>
> Reviewer 6hYY was positive about the paper, highlighting its relevance and the novelty of benchmarking genomic foundation models under adversarial settings. However, they requested more biological interpretation and clarity on robustness trends.
>
>
> **Rebuttal actions:**
>
>
> * Expanded **biological analysis of adversarial perturbations**, showing clustering in regulatory motifs and GC-content preservation.
> * Added **interpretability experiments** linking attack success to motif disruption.
> * Clarified **robustness trends across tokenization schemes and quantization settings**.
> * Provided additional **visualizations (Appendix F)** to illustrate motif-level vulnerability.
>
>
> **Reviewer response:**
>
>
> > “Thank you for your response. I will maintain my positive score.”
>
> ---
>
> ## **Reviewer [ptyu](https://openreview.net/forum?id=XXNh0JfsDN&noteId=oDBBMlhtJY) (Rating: 8, Confidence: 3)**
>
>
> **Initial stance:**
>
>
> Reviewer ptyu strongly supported the paper, praising its novelty and practical relevance. They suggested adding experiments on cross-species robustness and uncertainty quantification for completeness.
>
>
> **Rebuttal actions:**
>
>
> * Added **cross-species robustness evaluation** (human → mouse transfer) to demonstrate generalization under distribution shifts.
> * Incorporated **Bayesian uncertainty analysis** to quantify confidence under adversarial perturbations.
> * Discussed implications for **safety-critical genomics applications** in healthcare and bioinformatics.
>
>
> **Reviewer response:**
>
>
> > “Thank you for the response and additional experiments. I'll keep my high score.”
>
>
> ---
>
>
> ## Notes on Reviewer nV6Z’s Score Adjustment
>
>
> We still note that Reviewer nV6Z expresses some reservations regarding the evaluation. Based on the reviewer’s own comments, the relatively lower score results from **limited familiarity with the field**. The reviewer voluntarily **increases the score from 2 to 4** and **lowers their confidence level from 2 to 1**.
>
>
> **Reasons for the adjustment include:**
>
>
> * Partial understanding of the significance of the benchmark paper.
> * Limited recognition of the considerable time and effort required to adapt existing methods to genomic foundation models.
> * Moderate appreciation of the practical relevance and importance of GFM applications.
>
>
> Given these points, we kindly invite the Chairs to review these clarifications as well, to help ensure a **balanced and accurate assessment**. We acknowledge that some concerns may stem from misunderstandings, and we are fully open to continued dialogue to clarify any remaining points.
>
>
> ---
>
>
> Thank you for your consideration.
>
>
> Authors

---

### Meta-Review · Area_Chair_1Sib · 2026-01-03

**Summary:**

Reviewers largely value the paper as a needed unified benchmark for adversarial attacks/defenses on genomic foundation models, but the decision hinges on whether the contribution is sufficiently novel/insightful beyond adapting existing attacks and whether the paper provides clear genomics-specific constraints/biological validity + actionable analysis (not just a tool/report). One reviewer was initially negative on novelty. The proposed method shows limited technical contribution as an adaptation of existing attacks/defenses rather than techniques tailored to this specific setting.

The current version does not yet reach the bar for acceptance due to insufficient technical novelty and limited depth of domain-specific insight. The additional analyses added during rebuttal improve the paper, but do not fully resolve the main concern that the work is primarily an integration of existing methods with incremental genomic adaptations, and the narrative still feels closer to a technical report than a research contribution with clearly articulated new principles for genomic adversarial robustness.

**Reviewer Concerns:**

Addressed in rebuttal: fixed the DSR formula issue; clarified rank interpretation; added sequence reversal to support biological relevance of attack hotspots; added random mutation baseline to separate augmentation vs true adversarial defense; clarified GenomeOcean adaptation + why generative attacks were excluded (ethics/biosecurity); added cross-species shift + uncertainty analysis.

Still outstanding: limited novelty remains for at least one reviewer; code availability/reproducibility messaging could be cleaner; generative-robustness evaluation remains largely out of scope (acknowledged as future work).

**Reviewer Scores:**

TVsB: 4 → 6 (likely would stay 6 with full discussion)
ptyu: 8 → 8 (unchanged)
nV6Z: 2 → 4 (likely stays 4; still not fully convinced on novelty)
6hYY: 6 → 6 (unchanged)

---

### Decision · Program_Chairs · 2026-01-26

Reject